# 3D Human Pose Estimation with Muscles

**Kevin Zhu**    **AliAsghar MohammadiNasrabadi**    **Alexander Wong**    **John McPhee**
University of Waterloo
{k79zhu, aa27moha, a28wong, mcphee}@uwaterloo.ca

## Abstract

We introduce MusclePose as an end-to-end learnable physics-infused 3D human pose estimator that incorporates muscle-dynamics modeling to infer human dynamics from monocular video. Current physics pose estimators aim to predict physically plausible poses by enforcing the underlying dynamics equations that govern motion. Since this is an underconstrained problem without force-annotated data, methods often estimate kinetics with external physics optimizers that may not be compatible with existing learning frameworks, or are too slow for real-time inference. While more recent methods use a regression-based approach to overcome these issues, the estimated kinetics can be seen as auxiliary predictions, and may not be physically plausible. To this end, we build on existing regression-based approaches, and aim to improve the biofidelity of kinetic inference with a multihypothesis approach — by inferring joint torques via Lagrange's equations and via muscle dynamics modeling with muscle torque generators. Furthermore, MusclePose predicts detailed human anthropometrics based on values from biomechanics studies, in contrast to existing physics pose estimators that construct their human models with shape primitives. We show that MusclePose is competitive with existing 3D pose estimators in positional accuracy, while also able to infer plausible human kinetics and muscle signals consistent with values from biomechanics studies, without requiring an external physics engine.

## 1 Introduction

3D human pose estimation (HPE) is a fundamental task in computer vision that involves the localization of 3D human joints from images, which allows the user to track human movement from videos, leading to a plethora of potential downstream applications. However, since many pose estimators are purely data-driven, the inferred motion is modeled implicitly, which may lead to physically impossible poses and movements.

Physics-based human pose estimation (PHPE) methods aim to mitigate these artifacts by enforcing the underlying dynamics equations that govern the kinematic state $\mathcal{K} = \{\boldsymbol{q}, \dot{\boldsymbol{q}}, \ddot{\boldsymbol{q}}\}$,

$$\mathfrak{M}(\boldsymbol{q}, \mathcal{A}) \cdot \ddot{\boldsymbol{q}} + \mathfrak{C}(\boldsymbol{q}, \dot{\boldsymbol{q}}, \mathcal{A}) = \boldsymbol{\tau}_q + \mathfrak{F} \tag{1}$$

where $\boldsymbol{q}$ are generalized coordinates that describe motion, often in terms of translational (*e.g.* 3D position of the root) and rotational (*e.g.* joint rotations) degrees of freedom (DoF). We denote "dot" ($\dot{}$) as the time derivative and "double dot" ($\ddot{}$) as the 2nd time derivative of a variable. $\mathfrak{M}$ is the mass matrix and $\mathfrak{C}$ contains the Coriolis, centrifugal, and gravitational forces, for a human with anthropometric features $\mathcal{A}$ at a given state $\mathcal{K}$. Here, we loosely lump together a human's dimensions, mass and inertia properties, and other intrinsic and mobility features using the anthropometrics term $\mathcal{A}$. On the right hand side, $\boldsymbol{\tau}_q$ describes the human joint torques generated by each DoF, and $\mathfrak{F}$ are external forces, both in the generalized space.

In this paper, we deal with monocular pose estimation, where our only input source is a monocular video, without force sensors. When only one unknown external force $\mathfrak{F}$ is applied on the human,

39th Conference on Neural Information Processing Systems (NeurIPS 2025).

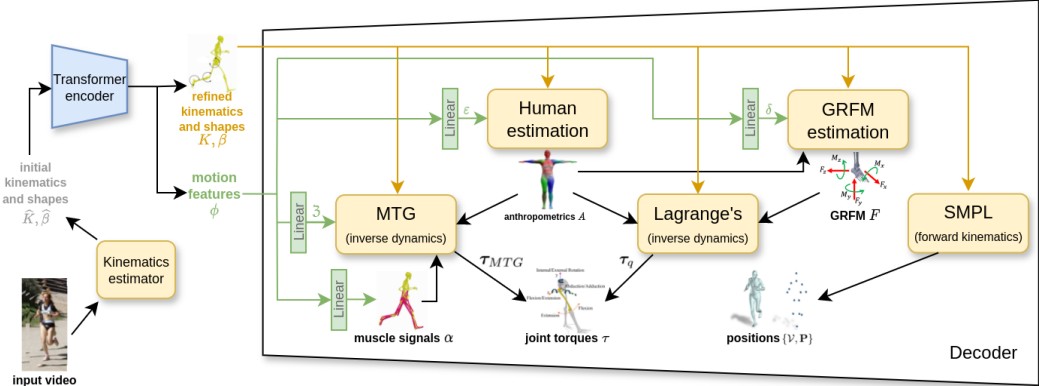

Figure 1: Overall framework of MusclePose.

we can solve for $\tau_q$ by enforcing the entries of $\tau_q$ that corresponds to the root link to be zero. However, when there are more than one external force applied to the human simultaneously at different locations, which is often the case (*e.g.* when both feet are in contact with the ground), Eq. (1) becomes underconstrained.

Without large-scale force-annotated video datasets, many methods estimate the corresponding kinetics via optimization with an external physics engine [20, 19, 77, 41]. However, these physics engines are often either non-differentiable and cannot be trained end-to-end, or are too slow for real-time inference. Furthermore, as discussed in [85], these methods are often combined with reinforcement learning to reach a desired outcome, but the effects of changing inputs on the outputs are unknown. Since joint torques can be hard to agree on in the biomechanics community, as they are often computed from different models, with different assumptions and post-processing, a more flexible learning framework may be preferred. More recently, PHPE methods have began regressing kinetics directly with neural networks [85, 37, 63]. While the regression-based approach improves the kinematic reliability of the predicted motion, the inferred kinetics can be seen as auxiliary predictions, which may not be directly constrained and may be physically implausible. Although these kinetic predictions are not the main focus of these pose estimators, they may still be of interest for downstream applications. In sports for example, in addition to kinematics, practitioners and researchers are often interested in analyzing the whole-body musculoskeletal dynamics of athletes. To do so, a multibody model of skeletal dynamics is commonly used in combination with an optimal control algorithm to generate predictive simulations of athlete movements [5, 28, 49]. However, these optimal control algorithms can take hours or days to produce results.

To this end, we build on existing regression-based PHPE approaches, to infer human kinetics simultaneously with kinematics, without a physics engine, and propose MusclePose (Fig. 1) to improve the plausibility of the predicted kinetics. To mitigate the underconstrained problem of regressing kinetics, we use a multihypothesis approach, and compute torques via Lagrange's equations, and also via muscle dynamics modeling with muscle torque generators (MTGs) [51, 25].

To maintain fidelity when modeling human movement, classical muscle models often represent muscles as linear actuators, and capture the nonlinear dependence of muscle tension on muscle length and the rate of lengthening [66, 32] using various Hill-type muscle models [23]. However, incorporating detailed muscles requires solving the actuator redundancy problem [3] and computing complex and varying musculoskeletal geometries [60, 12]. To overcome these drawbacks, parametric MTG models were proposed to mimic the behavior of muscles crossing a given joint to directly approximate joint torque by modeling kinematic dependence on active torque generation and passive impedance (Eq. (11)). Essentially, MTGs infer net joint torques from a joint's kinematics and activation levels, which is what we ultimately want, as we are not interested in isolated muscle tensions or granular joint contact forces. And since MTGs consist of differentiable equations, we are able to incorporate them into our learning framework, and train our pose estimator end-to-end.

Moreover, for computational efficiency, existing PHPE methods rely on human models with anthropometrics estimated from the predicted human dimensions, or use the intrinsics properties (*e.g.* inertia

and mass properties) of primitive shapes (*e.g.* spheres and simple rods), as proxies. From, Eq. (1), we see that, even if the kinematics state $\mathcal{K}$ and external forces $\mathfrak{F}$ are accurate, but $\mathcal{A}$ is not, the inferred torques $\boldsymbol{\tau}_q$ may not correspond to the actual human performing the motion. For example, existing pose estimators may infer the center of mass (CoM) of body parts by taking the mean of the predicted surface mesh, assuming constant density [85]. However, since the composition of bones, muscles, internal organs, etc. is different, the human body's density is not uniform [14]. For example, the CoM of the upper torso is slightly towards the left side [16], whereas taking the mean vertices will be in the center. As such, we further predict detailed anthropometrics for each human, and keep them close to values taken from biomechanics studies.

In summary, we introduce MusclePose to comprehensively predict human kinematics, kinetics, muscle signals, and detailed anthropometrics from monocular video. Specifically, we want a pose estimator with (i) a flexible learning framework easily adaptable for different scenarios, (ii) a reasonable degree of biofidelity, (iii) inference speed and (iv) positional accuracy both on par with purely kinematic pose estimators. To satisfy (i) and (iii), MusclePose is regression-based, consists of customizable and swappable components, can be trained end-to-end, and does not require an external physics engine. For (ii), MusclePose is the first pose estimator to incorporate muscle dynamics modeling and predict detailed human anthropometrics. We demonstrate improvements in the inferred kinetics on actions including *walking* from the H36M dataset [27] and *baseball pitching* and *golf swings* from PennAction [84]. Also, the use of MTGs allows us to further assess human motion at a musculoskeletal level, and we show that our inferred muscle signals are comparative to those from biomechanics studies, as well as to EMG data of pertinent muscle groups. Lastly, for (iv), we evaluate our method on benchmark 3D HPE datasets, H36M [27] and 3DPWoc, [71], to show that MusclePose is kinematically competitive with state-of-the-art (SOTA) pose estimators.

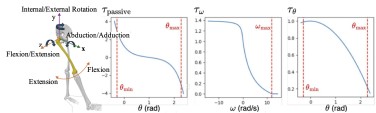

Figure 2: Examples of MTG curves for hip flexion. $\tau_{passive}$ models the passive torque [79] as a double exponential function. $\tau_{\omega}$ models the active-torque–angular-speed relationship [69, 67] as a piecewise function. $\tau_{\theta}$ models the active-torque-angle relationship [21, 33] as the non-negative portion of a polynomial.

## 2 Related work

**Monocular 3D human pose estimation.** Early deep learning 3D HPE approaches use convolutional neural networks to directly estimate human 3D keypoint positions from images, with intermediate values represented by 3D heatmaps [57], location maps [50], or 2D heatmaps with depth regression [87]. The more recent and popular approach lifts 2D keypoints to 3D, essentially forming a monocular sparse-depth estimation task. The lifting network can be fully-connected layers [47], temporal convolution networks [10, 58], graph convolution networks [74, 7, 11], or transformers [89, 39, 86]. Human pose and shape estimation (HPSE) refers to predicting a 3D surface mesh of humans. The popular model-based HPSE approach [38, 35, 31] predicts input parameters of a parametric human model, such as SMPL, which infers a 3D mesh from rotation and shape parameters. Non-parametric approaches [42, 52, 36] directly regress 3D coordinates of mesh vertices. Other methods combine both approaches, such as [70], which predicts a volumetric representation before fitting a SMPL model, or [45], which calibrates model-based mesh predictions with 3D keypoints.

**Physics based pose estimation.** To achieve more physically plausible human motion, PHPE methods apply dynamic constraints to encourage contact and penalize motion jitter, ground penetration, and unbalanced postures. [77, 63, 61, 64] model contact forces between the foot and ground, [20, 19, 82] include contact points between the full body and ground, while [41] also models human interaction with stick-like hand tools. Optimization-based frameworks [20, 19, 77, 64, 41] simulate physically plausible human motion from a physics engine and minimize an objective function to keep the simulated motion close to the detections obtained from a kinematic pose estimator. These frameworks are also combined with reinforcement learning [82, 59]. Recently, to overcome the need of an external physics engine, regression-based frameworks [85, 37, 63] directly estimate human kinetics using neural networks. As discussed in Sec. 1, this is an underconstrained problem, for which we hope to mitigate, and further inject biofidelity. We incorporate MTGs to do so.

**Muscle torque generators (MTGs).** Due to their simplicity, MTGs have been increasingly popular in multibody dynamics simulations as they reduce computational cost while maintaining a reasonable degree of biofidelity. Recently, MTGs have been incorporated to simulate human movement post hip and knee replacement surgeries [13], human interactions with exoskeletons [22, 26], and manual wheelchair propulsion [5]. For more dynamic movements, such as in sports, examples of MTG-driven simulations of athlete motor control include golf [49] and cycling [28].

## 3    MusclePose

We propose MusclePose (Fig. 1) as a physics-based pose estimator to directly regress comprehensive human dynamics from a monocular video of length $T$. We use a transformer encoder to refine initial pose estimates, and produce latent motion features $\phi^{\{1:T\}}$, which are used as inputs for 5 customizable modules to infer human anthropometrics $\mathcal{A}$, kinematics $\mathcal{K}$, external forces $\mathcal{F}$, joint torques via Lagrange's equations $\boldsymbol{\tau}_q$, and joint torques via MTGs $\boldsymbol{\tau}_{MTG}$. We describe our prototype in the following subsections, where we extract muscle signals $\boldsymbol{\alpha}^{\{1:T\}}$ and residual terms $\mathcal{E}, \delta^{\{1:T\}}$, $\mathfrak{Z}$ from $\phi^{\{1:T\}}$ as inputs for the 5 modules. All variables described in this section are sequences of length $T$, except $\mathcal{E}, \mathfrak{Z}$ and shape parameters $\boldsymbol{\beta}$, and we drop the superscript $(^{\{1:T\}})$.

Compared to the most recent regression-based PHPE method, PhysPT [85], we do not use a transformer decoder to directly regress joint torques, and instead, regress the input parameters of MTG models. Our motivation stems from the popular HPSE approach that regresses SMPL parameters instead of the human mesh directly, which not only reduces the computation complexity but also geometrically constrains the predicted human, as SMPL infers the mesh via forward kinematics. In parallel, we use MTGs to avoid estimating complex musculoskeletal geometries and granular joint contact forces, while enforcing a constraint on the inferred torques from Lagrange's equations.

### 3.1    Kinematics estimation

We follow the common approach in PHPE and clean initial kinematic estimates $\{\hat{\boldsymbol{\theta}}, \hat{\boldsymbol{\beta}}, \hat{\mathbf{T}}\}$ generated by some existing kinematic pose estimator, to obtain the refined $\{\boldsymbol{\theta}, \boldsymbol{\beta}, \mathbf{T}, \mathbf{c}\}$ as our prediction. Here, $\mathbf{T} \in \mathbb{R}^3$ represents 3D pelvis **translation** in the world frame, and $\mathbf{c}$ are binary **contact** labels. **Rotation** parameters $\boldsymbol{\theta} = \{\boldsymbol{\theta}_0, ..., \boldsymbol{\theta}_{23}\}$ represents local rotations of the 24 SMPL keypoints, relative to their parents in the SMPL kinematic tree, with $\boldsymbol{\theta}_0$ being the pelvis orientation in the world frame. We follow prior work [34] and predict the 6D continuous rotation representation [88] for each $\boldsymbol{\theta}_k \in \mathbb{R}^6$. **Shape** parameter $\boldsymbol{\beta} \in \mathbb{R}^{10}$ denotes the first 10 principal components of SMPL's shape space. Since our inputs lack the shape information that RGB images provide, we follow the hybrid approach in [89] to regress shape residuals that are combined with initial predictions. We also use the same approach and regress anthropometric residuals $\mathcal{E}$ later on in Sec. 3.2 and force residuals $\delta$ in Sec. 3.3.

The parametric human model, SMPL, then uses a collection of linear functions to map these parameters to a triangulated mesh $\mathcal{V}$ of 6890 vertices that represents the surface of the human body, and 24 SMPL keypoint positions $\mathbf{P}$:

$$\{\mathcal{V}, \mathbf{P}\} = \text{SMPL}(\boldsymbol{\theta}, \boldsymbol{\beta}) + \mathbf{T} \tag{2}$$

We define the kinematic loss $\mathcal{L}_{kin}$ with weights $\boldsymbol{\lambda}_{kin}$ as

$$\mathcal{L}_{kin} = \boldsymbol{\lambda}_{kin} \cdot [\mathcal{L}_p \quad \mathcal{L}_v \quad \mathcal{L}_\theta \quad \mathcal{L}_\beta \quad \mathcal{L}_{norm} \quad \mathcal{L}_c]^\mathsf{T} \tag{3}$$

where the first five losses are from [89] which penalize joint position, linear velocity, SMPL parameter prediction L1 errors, and minimize the L2 norms of the SMPL parameters; and $\mathcal{L}_c$ is the binary contact loss from [83].

To facilitate multibody dynamics modeling in the following sections, we convert the predicted coordinates to **generalized coordinates**, $\boldsymbol{q} = [\boldsymbol{X}_0, \boldsymbol{q}_0, \boldsymbol{q}_1, ..., \boldsymbol{q}_{N_k}]^\mathsf{T} \in \mathbb{R}^{N_{DoF}}$, where $\boldsymbol{X}_0 \in \mathbb{R}^3$ is the global root translation, and each $\boldsymbol{q}_k$ describes the joint's rotational DoFs. Specifically, each $q_i \in \boldsymbol{q}_k$ are ZXY euler angles converted from the predicted $\boldsymbol{\theta}_k$, to match the International Society of Biomechanics (ISB) format, where a joint's local $z$-direction corresponds to flexion/extension, $x$ for abduction/adduction and $y$ for internal/external rotation. We denote the predicted kinematics as $\mathcal{K} = \{\boldsymbol{q}, \dot{\boldsymbol{q}}, \ddot{\boldsymbol{q}}\}$, with the velocity and acceleration terms estimated via finite differences.

## 3.2 Human estimation

**Human model.** We assume a rigid multibody dynamics model of a human with $N_k = 18$ segments and $N_{DoF} = 47$ total degrees of freedom (DoF). The 3D positions of the 18 joints, each corresponding to a segment, are the 24 SMPL keypoint positions minus the 5 end-effectors and the *spine3* SMPL keypoint. The wrists, elbows, scapulas, each contain 2 rotational DoFs, knees each with 1 rotational DoF, root with 3 rotational and 3 translational DoFs, and 3 rotational DoFs for each of the remaining joints, for a total of 47 DoFs. We selected this configuration as it aligns best with biomechanics studies with anthropometric measurements that we use in the remaining sections.

**Anthropometrics prediction.** To predict the human's anthropometrics $\mathcal{A} = \cup_k \{m_k, I_{0,k}, CoM_k\}$, specifically the **mass** $m_k$, **inertia tensor** at zero rotation $I_{0,k}$, and **CoM** of all segments, we scale literature values $\bar{\mathcal{A}}$ from [16] based on the predicted human shapes $\beta$ and add the predicted offsets $\mathcal{E}$,

$$\mathcal{A} = s_\beta \bar{\mathcal{A}} + \mathcal{E} \tag{4}$$

The scaling term $s_\beta$ is computed from the predicted $\beta$, with details in the supplementary material.

## 3.3 Kinetics estimation

**Ground reaction forces and moments (GRFM) prediction.** Let $\mathcal{F}_k = [\mathbf{F}_k, \mathbf{M}_k]^\intercal$ be the GRFM applied on the CoM of each segment $k$. We infer $\mathcal{F} = \sum_k \mathcal{F}_k$ from our previous predictions and our regressed force residuals $\delta$,

$$\mathcal{F} = \text{GRFM model}(\mathcal{K}, \mathcal{A}, \delta) \tag{5}$$

Since we trained our model on the AMASS dataset [46] and feet-ground contact labels from RoHM [83], we assume feet-ground contact only for simplicity, as with many PHPE methods [37, 20, 82]. Omitting subscript $k$, let $\mathbf{F} = [F_X, F_Y, F_Z]^\intercal$ be the force in world cartesian coordinates where $Y$ is the vertical direction, and let $\mathbf{z} = [z_x, z_y, z_z]^\intercal$ be the center of pressure (CoP) in the foot's local coordinates where $x$ is along the length of the foot (*i.e.* $\mathbf{M} = R^0_{ankle} \mathbf{z} \times \mathbf{F}$ where $R^0_{ankle}$ is the ankle's world orientation). From the regressed residuals $\delta_{\{Y,l\}} \subset \delta$ and the kinematics of each foot $\mathcal{K}_{foot}$, we estimate the vertical force applied on the foot scaled by bodyweight $F_Y^W = F_Y/W$, and the CoP along the foot scaled by foot length $z_x^l = z_x/l_{foot}$,

$$\{F_Y^W, z_x^l\} = \eta \mathcal{K}_{foot} + \delta_{\{Y,l\}} \tag{6}$$

where linear coefficients $\eta$ were fitted on the forceplate data in [72]. The remaining $\delta$ terms are scaling factors between -1 and 1 to ensure the values in the other directions are physically possible (*i.e.* $F_X^2 + F_Z^2 \leq \delta_\mu^2 F_Y^2$ and $\mathbf{z}$ is within the foot's dimensions)

$$F_X = \delta_X \delta_\mu F_Y, \qquad\qquad F_Z = \delta_Z \sqrt{\delta_\mu^2 F_Y^2 - F_X^2} \tag{7}$$

$$z_y = -|\delta_h l_h|, \qquad\qquad z_z = \delta_s(l_w/2) \tag{8}$$

where $l_w, l_h$ are the foot's width and height, respectively. Additional details of our GRFM model can be found in the supplementary material.

**Inverse dynamics via Lagrange's.** From here, we can analytically compute the mass matrix $\mathfrak{M}$, Coriolis term $\mathfrak{C}$, external forces in the generalized space $\mathfrak{F}$, and infer joint torques in the generalized space $\boldsymbol{\tau}_q$ from the equations of motion:

$$\boldsymbol{\tau}_q = \text{Lagrange's}(\mathcal{K}, \mathcal{A}, \mathcal{F}) = \mathfrak{M}\ddot{\boldsymbol{q}} + \mathfrak{C} - \mathfrak{F} \tag{9}$$

We include the calculations of these terms in the supplementary material. We define a residual force loss $\mathcal{L}_{res}$ to minimize the resulting forces and torques at the root, which correspond to the first 6 entries of $\boldsymbol{\tau}_q$

$$\mathcal{L}_{res} = |\boldsymbol{\tau}_{q_{[:6]}}| \tag{10}$$

**Inverse dynamics via MTGs.** Simultaneously, we use parametric MTG models [51] to infer joint torques $\boldsymbol{\tau}_{MTG}$ from the predicted kinematics $\mathcal{K}$ and muscle activations $\boldsymbol{\alpha}$. Specifically, this kinematic dependence is separated into active torque generation $\tau_{active}$ and passive impedance $\tau_{passive}$. For each joint rotational DoF $q \in \boldsymbol{q}_{[6:]}$ with angular velocity $\dot{q}$, let muscle signal $\alpha \in [0, 1]$ represent the joint's corresponding activation level for this DoF, we compute the corresponding torque as

$$\tau_{MTG} = \text{MTG}(\mathcal{K}, \mathcal{A}, \alpha, \mathfrak{Z}) = \tau_{active} + \tau_{passive} \tag{11}$$

The **active torque** is further broken down into

$$\tau_{active} = \alpha \cdot \tau_\omega(\dot{q}) \cdot \tau_\theta(q) \cdot \tau_0(\mathcal{A}, \mathfrak{Z}) \tag{12}$$

where $\tau_\omega(\dot{q}; \boldsymbol{\gamma}_\omega)$ models the active-torque–angular-speed relationship [69, 67] and $\tau_\theta(q; \boldsymbol{\gamma}_\theta)$ models the active-torque-angle relationship [21, 33], as shown in Fig. 2. These relationships are parameterized by the $\boldsymbol{\gamma}$ coefficients, which are unique for each joint's DoF and direction, and are identified via dynamometry. This joint-dependent parameterization preserves physiological realism (*e.g.*, hip flexion and knee extension should exhibit different peak torque and passive stiffness profiles), unlike uniform torque models that assume identical properties across the body. For this paper, we use the set of $\boldsymbol{\gamma}$ values summarized in [54, 53].

$\tau_0(\mathcal{A}; \boldsymbol{\gamma}_i, \boldsymbol{\gamma}_e)$ is the peak isokinetic torque that controls peak MTG output at zero joint velocity, which can be measured with a dynamometer. $\tau_0$ is estimated in [54, 53] as a linear approximation of the human's intrinsic, scaled by certain external factors such as the human's fitness or activity level. Since these external factors (and some intrinsic properties) are not readily known, we take the mean effects $\boldsymbol{\gamma}_i, \boldsymbol{\gamma}_e$ from [54, 53], and add regressed offsets $\mathfrak{Z}$. We compute $\tau_0$ as

$$\tau_0 = (\boldsymbol{\gamma}_i \mathcal{A} + \mathfrak{Z}_i)(\boldsymbol{\gamma}_e + \mathfrak{Z}_e) \tag{13}$$

Furthermore, to account for stability, we assume each joint is driven by a pair of agonist-antagonist MTGs — a flexor (+) and an extensor (-), that corresponds to the movement direction. Hence, for each joint rotational DoF, we regress 2 muscle signals $\{\alpha^{flex}, \alpha^{ext}\}$, and the active torque becomes:

$$\tau_{active} = \alpha^{flex}\tau_\omega^{flex}\tau_\theta^{flex}\tau_0^{flex} + \alpha^{ext}\tau_\omega^{ext}\tau_\theta^{ext}\tau_0^{ext} \tag{14}$$

$\tau_{passive}(q; \boldsymbol{\gamma}_p)$ is the **passive torque** [1] of a joint that arises when the surrounding muscles, tendons, and ligaments are strained and intensifies near anatomical joint limits [1, 81]. The joint's viscous damping and nonlinear stiffness are parameterized by $\boldsymbol{\gamma}_p$, which encourages the joint to move within its range of motion, as a large restoring torque is produced otherwise, as shown in Fig. 2. Equations to compute $\tau_\omega, \tau_\theta, \tau_{passive}$ can be found in the supplementary material.

We define the torque loss $\mathcal{L}_\tau$ as the absolute difference between the two sets of predicted joint torques, and another regularizing term $\mathcal{L}_\epsilon$ for all regressed residuals:

$$\mathcal{L}_\tau = |\boldsymbol{\tau}_{q[6:]} - \boldsymbol{\tau}_{MTG}| \tag{15}$$

$$\mathcal{L}_\epsilon = ||\mathcal{E}||_2 + ||\delta||_2 + ||\mathfrak{Z}||_2 \tag{16}$$

Finally, we have dynamic loss with weights $\boldsymbol{\lambda}_{dyn}$

$$\mathcal{L}_{dyn} = \boldsymbol{\lambda}_{dyn} \cdot [\mathcal{L}_\tau \quad \mathcal{L}_{res} \quad \mathcal{L}_\epsilon]^\mathsf{T} \tag{17}$$

## 4 Experiments

### 4.1 Implementation and datasets

For training, we used the AMASS dataset [46], with feet-ground contact labels from [83]. As such, we trained and evaluated on sequences with feet-ground contact only (denoted $^\dagger$), which is also the case for many PHPE experiments [37, 20, 82]. We trained MusclePose end-to-end with a sequence input length of 16 frames, using total loss $\mathcal{L}_{total} = \mathcal{L}_{kin} + \mathcal{L}_{dyn}$ for 25 epochs, using the AdamW optimizer [44] with a weight decay of $10^{-4}$ and an initial learning rate of $10^{-4}$ that decreases by 20% every 5 epochs. Following common curriculum learning [2] practices, we split the training into two phases — for the first 20 epochs, we trained using the ground truth as input, followed by 5 epochs using the model's predictions as inputs.

For evaluation, we assessed positional accuracy on the inference results of the H36M test set [27] and object-occlusion subset of 3DPW (3DPWoc) [71]. As with training, we removed input sequences containing non feet-ground contact, the *sitting* and *sitting down* actions in H36M, and *courtyard laceshoe, flat guitar, outdoors climbing, outdoors freestyle, outdoors parcours, downtown stairs* in 3DPWoc. We further assessed kinetic biofidelity from 3 actions — *walking* from H36M, and *baseball pitch* and *golf swing* from the PennAction dataset (PA) [84]. As with existing large-scale human video datasets, since neither datasets include force-annotations, we compared our inference results with existing biomechanics studies of these movements, and commented on overall trends and plausibility. We selected *walking* because human gait is heavily studied in biomechanics [72, 18, 76, 8, 29, 75], and is a relatively consistent and cyclic movement. We included the latter two actions to evaluate faster movements, for which we were able to find published lab measurements [55, 80, 62]. During inference, to promote a closer comparison with the SOTA regression-based physics pose estimator PhysPT [85], we used the same kinematic estimator, CLIFF [40], to extract initial kinematic estimates, and the global trajectory predictor in [85] to extract initial root DoFs. The rationale for using CLIFF in [85] is that it produces competitive positional accuracy but lacks in physical plausibility.

## 4.2 Positional accuracy

We followed standard evaluation protocol and reported the mean per-joint positional error (MJE) and procrustes-aligned MJE (PJE) in Tab. 1, for the 14 LSP [30] keypoints in millimetres. MJE is the root-aligned mean Euclidean distance in millimeters between the predicted and ground truth 3D keypoints. PJE is the MJE after aligning the predicted pose with the ground truth in translation, rotation, and scale using the Procrustes method. For 3DPWoc, since the data is captured using a moving camera with unknown extrinsics, and our method predicts the global root DoFs directly, we reported PJE only, with additional ablations in Tab. 3 to show consistency of results.

We see that MusclePose outperforms other PHPE methods on H36M but is slightly worse than PhysPT on 3DPWoc. This, along with the overall worse positional accuracy of PHPE compared to purely kinematic methods, could be due to a kinematics-kinetics trade-off, as our method produced a lower residual force (Tab. 4). Specifically, the root DoFs in 3DPW may be harder to estimate due to the moving camera, leading to higher residual forces, which the model may try to reduce (lower kinetic error) by estimating a set of local joint kinematics slightly different from the original motion (higher kinematic error).

Table 1: Positional accuracy on H36M and 3DPWoc. "opti" denotes kinetics obtained from an external physics optimizer, and "regr" denotes regressed by a neural network. [†]denotes sequences with feet-ground contact only.

|  |  | | H36M | | 3DPWoc |
|  |  | Kinetics | MJE↓ | PJE↓ | PJE↓ |
| --- | --- | --- | --- | --- | --- |
| kinematic | HybrIK [38] | - | 55.4 | 33.6 | - |
|  | HybrIK [38] | - | [†]56.4 | [†]36.7 | - |
|  | CLIFF [40] | - | 52.2 | 36.8 | - |
|  | CLIFF [40] | - | [†]**46.5** | [†]**32.4** | [†]**24.0** |
| PHPE | SimPoE [82] | opti. | [†]56.7 | [†]41.6 | - |
|  | DiffPhy [20] | opti. | [†]81.7 | [†]55.6 | - |
|  | D&D [37] | regr. | [†]52.5 | [†]35.5 | - |
|  | PhysPT [85] | regr. | [†]50.6 | [†]35.5 | [†]25.9 |
| | MusclePose(ours) | regr. | [†]48.4 | [†]33.5 | [†]27.6 |

## 4.3 Biofidelity

Since most PHPE methods do not report kinetics results, we mainly compared ours with values we reproduced from PhysPT. For even comparison, for both pose estimators, we computed joint torques in the generalized space $\tau_q$ via Lagrange's equations (9) from the predicted motion, anthropometrics, and GRFM. Unlike the biomechanics studies, we did not apply additional signal post-processing or smoothing to $\tau_q$. Hence, we see more noise and spikes in the pose estimators' results, which could also be amplified by the low frame rate of PennAction.

**Qualitative.** Since the different biomechanics studies computed torques differently from different datasets, we comment on general trends and evaluate qualitatively. In Fig. 3, we plotted median torques scaled by predicted body weight, with a 25-75% quantile band, for select joints of the 3 actions, for which we found reference values. Overall, compared to PhysPT (gray), we see that MusclePose (ours, purple) more closely follows the trends and magnitudes of the reference values (greens and yellow).

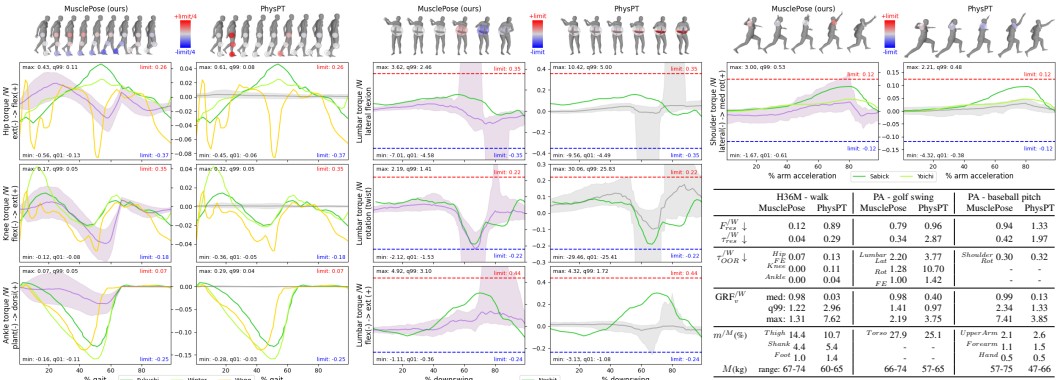

Figure 3: Median predicted joint torques scaled by bodyweight, with a 25-75% quantile band, for gait cycles, the downswing phase of golf drives, and the arm acceleration phase of baseball pitches, compared to values from biomechanics studies.

The 2 leftmost columns of Fig. 3 correspond to flexion/extension of the hip, knee, and ankle torques for walking, scaled such that the toe-off occurs at 60% of the gait cycle, and compared to reference torques from [73, 18, 75]. We see that MusclePose produced more reasonable trends overall, whereas PhysPT produced torques with low magnitudes but with higher extreme values. For hip flexion, our results resembled more of the yellow curve that was computed from a wearable system in [73], where the authors attributed their errors to a lack of shear force measurement, leading to more noticeable errors in the hips than more distal joints due to the increase in moment arms. Since the subjects in H36M walk in a small circle with slower and varying speeds, a discrepancy can arise in the generated shear force, leading to the discrepancy in hip torque, as the subjects in the biomechanics studies walk in a straight line at a consistent pace. This could also contribute to the low magnitude of ankle torque, where different timings (toe-off/heel-off/touch-down) could affect ankle power generation, as explained in [6].

The middle 2 columns of Fig. 3 correspond to lumbar torques during the downswing phase of golf drives, scaled such that the maximum lumbar rotation occurs at 2/3 of the motion, and compared to reference torques from [55]. While we see noticeable spikes from both pose estimators, the extreme values for lumbar lateral bending and axial rotation were much higher for PhysPT.

The 2 rightmost columns of Fig. 3 correspond to shoulder lateral/medial rotation torque during the arm acceleration phase of baseball pitches, scaled such that the maximum moment occurs at 80% of the motion, and compared to reference torques from professional pitchers (darker green) in [62], as well as amateurs (lighter green) in [80]. Although the peak of our 75% quantile slightly exceeds the shoulder medial rotation limit reported in [62], our band was able to cover values from both skill groups, whereas PhysPT's band was below the amateurs, even though the pitchers in PA range from teenage amateurs to adult professionals.

**Quantitative.** In the bottom right of Fig. 3, we reported the mean residual forces $\{F_{res}, \tau_{res}\}$, mean out of range joint torques $\tau_{OOR}$, and the median value of the sum of GRFs in the direction opposite of gravity $\text{GRF}_v$. Residual $F_{res}$ and $\tau_{res}$ were computed as the mean L2 norms of the entries of $\boldsymbol{\tau}_q$ that correspond to the translational and rotational DoFs of the root. $\tau_{OOR}$ was computed as the mean absolute amount outside of joint torque limits (red and blue values in Fig. 3) reported in [1, 43, 62]. These values are further scaled by predicted body weight and denoted ($^{/W}$). In the last column, we also reported the mean predicted segment mass $m^{/M}$ as a percentage of body mass $M$.

Overall, we see that MusclePose inferred more reasonable kinetics, indicated by the lower residual forces, less extreme joint torques, and a median $\text{GRF}_v$ closer to body weight. Similar to its joint torques, PhysPT's $\text{GRF}_v$ values were overall lower in magnitude, with occasional spikes. While MusclePose's maximum $\text{GRF}_v$ were very large for the golf swing and baseball pitch, our 99% quantiles were more comparable to the maximum values of about 1.3 times body weight for golf reported in [55], and about 2.3 for pitching in [9].

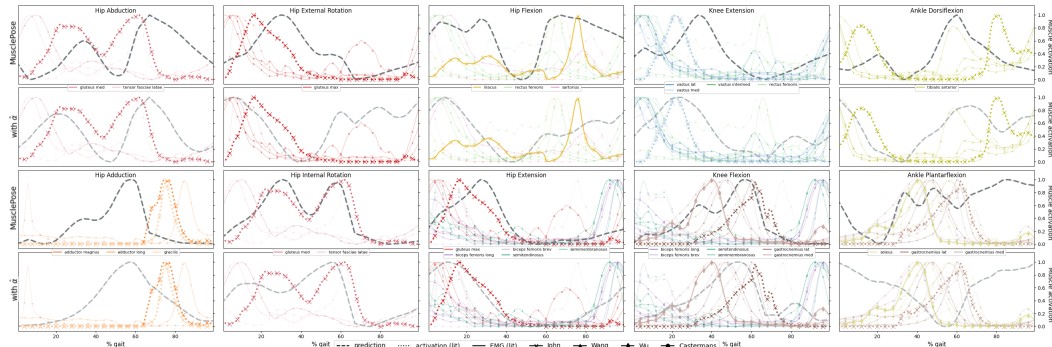

Figure 4: Mean predicted muscle activations (dashed) for gait cycles compared to EMG data and predictions of pertinent muscles from literature. Min-max scaling applied to all values.

## 4.4 Ablations and additional evaluation

We reported ablations results in Tab. 2. Row 2 includes results without custom anthropometrics, and instead computed directly from the SMPL mesh, assuming constant density. Row 3 includes results without MTGs. Row 4 includes results using kinematic-based muscle activations, with details in the next paragraph. We see that, overall, MusclePose has better positional accuracy, along with lower residual forces ($\mathbf{F}_{res} = (F_{res} + \tau_{res})/2$), and median $\text{GRF}_v$ closest to bodyweight.

Table 2: Ablations results.

| | [†]H36M | | [†]3DPWoc | | H36M - Walk | | PA - Golf swing | | PA - Pitch | |
|---|---|---|---|---|---|---|---|---|---|---|
| | MJE↓ | $\mathbf{F}_{res}^{/W}$ ↓ | PJE↓ | $\mathbf{F}_{res}^{/W}$ ↓ | $\mathbf{F}_{res}^{/W}$ ↓ | med $\text{GRF}_v^{/W}$ | $\mathbf{F}_{res}^{/W}$ ↓ | med $\text{GRF}_v^{/W}$ | $\mathbf{F}_{res}^{/W}$ ↓ | med $\text{GRF}_v^{/W}$ |
| MusclePose | **48.4** | **0.08** | 27.6 | **0.25** | **0.08** | **0.98** | **0.56** | **0.98** | 0.68 | **0.99** |
| w/o $\mathcal{A}$ | 49.5 | 0.26 | 27.9 | 0.35 | 0.22 | 0.74 | 0.61 | 0.64 | **0.67** | 0.39 |
| w/o $\tau_{MTG}$ | 49.7 | 0.14 | **27.2** | 0.30 | 0.15 | 0.94 | 0.66 | 0.88 | 0.74 | 0.90 |
| with $\hat{\alpha}$ | 50.9 | 0.16 | 28.4 | 0.26 | 0.13 | 0.93 | 0.59 | 0.97 | 0.69 | 0.90 |

Furthermore, due to the lack of extrinsics information in 3DPWoc, we reported results from using different kinematic estimators in Tab. 3 to show consistency.

Table 3: [†]3DPWoc results with different kinematic estimators.

| | Ours+CLIFF | CLIFF [40] | Ours+WHAM | WHAM [65] | Ours+CoMotion | CoMotion [56] |
|---|---|---|---|---|---|---|
| PJE↓ | 27.6 | 24.0 | 28.5 | 22.7 | 32.6 | 30.7 |
| $\mathbf{F}_{res}^{/W}$ ↓ | 0.3 | - | 0.3 | - | 0.2 | - |

**Muscle activations.** Since we were not able to find public video datasets with corresponding MTG activation signals to directly compare with, we followed the evaluation procedure in [29] to assess general trends, and overlayed our mean muscle activation predictions (black, dashed) for gait cycles from H36M with EMG data of pertinent muscles from other gait studies [72, 76, 8, 29] in rows 1 and 3 of Fig. 4, with min-max scaling applied to all values. We also included the predicted activations (dotted) from [29]; however, they use a different muscle model. While muscle activations can be seen as surrogate representations of EMGs, the two are not exactly the same. Hence, mismatches in timing and magnitude will exist, and peaks and valleys may be further amplified by the min-max scaling applied. In general, raw EMG values can vary widely due to electrode placement.

To mimic methods that regress joint torques as a linear combination of joint kinematics, we experimented with estimating the muscle activation as a "kinematic effort term" (denoted $\hat{\alpha}$), specifically as a joint's angular velocity relative to its limit plus an additionally regressed offset term:

$$\hat{\alpha}^d = \dot{q}^d/\dot{q}_{max}^d + \mathcal{Z}_\alpha^d, \qquad d \in \{flex, ext\} \qquad (18)$$

We plotted $\hat{\alpha}$ results (gray, dashed) in rows 2 and 4 of Fig. 4. In comparison, MusclePose (rows 1 and 3) seems to better follow literature trends overall, such as having a more noticeable hitch (or second peak) for hip adduction, external rotation, knee flexion, etc. Furthermore, for the $\hat{\alpha}$ case, ankle

plantarflexion seems to be deactivated during the middle of the gait cycle, when it should peak. Row 4 of Tab. 2 also shows that MusclePose quantitatively outperforms the $\hat{\alpha}$ case.

**Kinematic plausibility.** In addition to the joint positional errors in Sec. 4.2, metrics such as acceleration loss (ACC), foot skating (FS), and ground penetration (GP) were introduced to further evaluate kinematic plausibility. We computed these values for H36M and 3DPWoc in Tab. 4, where ACC is the mean L2 norm in mm/frame$^2$ between the predicted and ground truth keypoint accelerations to access jitter. We also included mean torque variation (MTV) as the mean absolute change in joint torques over consecutive frames (in Newton*metres/frame) to assess torque continuity. FS is the average displacement in mm of vertices in contact with the ground in consecutive frames. GP is the average vertical distance to the ground in mm of vertices below the ground.

Table 4: Plausibility metrics. $^P$ indicates Procrustes aligned. (%f) indicates % of frames.

| | | Pos.
MJE↓ | Kinematic plausibility
ACC↓ | FS | GP | Float (%f)
$\mathcal{H}_{min} > \{1, 10, 20\}$mm | | | Kinetic plausibility
$\mathbf{F}_{res}^{/W}$↓ | MTV | GRF$_v^{/W}$
{med | q99 | max} | GRF$_v^{/W}$ (%f)
< {0.01, 0.1, 0.5} | | |
|---|---|---|---|---|---|---|---|---|---|---|---|---|---|---|---|---|
| **†H36M** | CLIFF | **46.5** | 26.3 | - | - | - | | - | - | - | - | | - | - | | - |
| | PhysPT | 50.6 | 13.7 | 34.7 | 6.8 | {59.0 | 31.6 | 8.5} | 0.4 | 5.3 | {0.4 | 2.4 | 10.0} | {7.2 | 20.8 | 60.0} |
| | MusclePose | 48.4 | **12.9** | 37.2 | 26.0 | {8.0 | 3.0 | 1.3} | **0.1** | 2.5 | {1.0 | 1.2 | 3.0} | {3.0 | 3.0 | 5.2} |
| **†3DPWoc** | CLIFF | **24.0**$^P$ | 13.8$^P$ | - | - | - | | - | - | - | - | | - | - | | - |
| | PhysPT | 25.9$^P$ | **3.0**$^P$ | 7.8 | 11.2 | {82.9 | 73.9 | 57.2} | 0.9 | 27.0 | {0.5 | 1.2 | 3.9} | {5.3 | 11.8 | 52.4} |
| | MusclePose | 27.6$^P$ | 4.3$^P$ | 12.8 | 30.8 | {6.0 | 4.7 | 3.7} | **0.3** | 12.1 | {1.0 | 1.6 | 4.3} | {5.3 | 5.3 | 6.3} |

While we see an improvement in jitter from both physics pose estimators, as indicated by a lower ACC compared to CLIFF, a lower FS or GP may not be strictly better. For one, foot sliding may occur naturally. And in terms of the latter, while human bodies deform under pressure and contact, the SMPL mesh does not model this deformation, and will instead penetrate the object it is in contact with [68]. During walking for example, minimizing GP while assuming this rigidity may restrict natural ankle rotation, potentially leading to the smaller ankle torques compared to literature values in Fig. 3. On the other hand, we should also check for floating. We reported the percentage of frames (%f) when the minimum vertex height $\mathcal{H}_{min}$ is above certain thresholds (1, 10, 20mm). Since we removed the non-feet-ground contact sequences, there are very minimal frames where "floating" occurs. We see that while PhysPT has less GP, it also includes more floating.

**Ground reaction force.** We can also characterize floating as when GRF$_v$ is small. As such, we also reported the percentage of frames when GRF$_v$ is below certain thresholds (1%, 10%, 50% of body weight) in Tab. 4. The results are consistent with $\mathcal{H}_{min}$, both indicating less floating (lower %f) for MusclePose. In Fig. 5, we plotted the predicted median vertical GRF of a foot, divided by body weight, for gait cycles in H36M. Compared to PhysPT (gray), we see that MusclePose's predictions (purple) are closer to literature values (greens) from [18, 75].

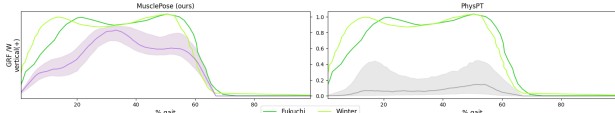

Figure 5: Median vertical GRF divided by body weight, of a foot for gait cycles in H36M, with a 25-75% quantile band.

## 5 Conclusion

In conclusion, we introduced MusclePose as the first PHPE method to simultaneously predict human kinematics, kinetics, muscle signals, and detailed anthropometrics from monocular video. In Sec. 4.3 and 4.4, we showed how the additions of muscle-dynamics modeling and detailed anthropometrics predictions improve the kinetic plausibility of regression-based PHPE, while being competitive with purely-kinematic pose estimators in positional accuracy in Sec. 4.2. Our framework consists of customizable components, does not require an external physics engine, and can be trained end-to-end.

**Acknowledgements.** We acknowledge financial support from the Canada Research Chairs Program, Canadian Sports Institute Ontario, and a Mitacs grant.

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

# Supplementary material for 3D Human Pose Estimation with Muscles

## A  Technical appendix

### A.1  Human model

We assume a rigid multibody dynamics model of a human with $N_k = 18$ joints – *pelvis*, *lumbar joint*, *thoracic joint*, *neck*, *scapulas*, *shoulders*, *elbows*, *wrists*, *hips*, *knees*, and *ankles*. The pelvis is set as the root, with 3 rotational and 3 translational degrees of freedom (DoFs). The scapulas have 2 DoFs, corresponding to depression/elevation and protraction/retraction. The elbows have 2 DoFs, corresponding to flexion/extension and forearm pronation/supination. The wrists have 2 DoFs, corresponding to flexion/extension and ulnar/radial deviation. The knees have 1 DoF, corresponding to flexion/extension. All remaining joints have 3 DoFs, for a total of 47 DoFs. We selected this configuration as it aligns best with existing biomechanics models that we implemented, such as for anthropometrics estimation [16] and MTGs [51, 25].

**Anthropometrics estimation.**   We predict the human's anthropometrics by combining our regressed residuals $\mathcal{E}$ with initial estimates based on literature values $\bar{\mathcal{A}}$ scaled by our predicted human dimensions. Specifically, we want to predict $\mathcal{A} = \cup_k \{m_k, I_{0,k}, \chi_k\}$, where $m_k$ is the **mass** of segment $k$, with $I_{0,k}$ as its **inertia** tensor at zero rotation with scaling matrix $\Lambda_k$, and $\chi_k$ as its local **CoM** position relative to its segment length. For the remainder of the section, we assume relevant units to be in seconds, radians, meters, kilograms, Newtons.

From the predicted $\boldsymbol{\beta}$ and Eq. (2), we can compute the human's volume and all segment lengths $L_k$. We further compute the human's initial bodymass estimate $\hat{M}$ as its volume multiplied by a constant density of $985 \ kg/m^2$. For segment $k$, let $s_{L,k} = L_k/H$ be its segment length relative to height, and $s_{m,k} = m_k/M$ be its mass relative to bodymass. Let "bar" ($\bar{\ }$) denote the human values measured by Dumas *et al.* in [16]. We set our initial estimates as $\bar{\mathcal{A}}$ scaled by $s_{L,k}/\bar{s}_{L,k}$:

$$\{M, s'_{m,k}, \Lambda_k, \chi_k\} = \{\hat{M}, \bar{s}_{m,k}\frac{s_{L,k}}{\bar{s}_{L,k}}, \bar{\Lambda}_k, \bar{\chi}_k\} + \mathcal{E} \tag{19}$$

$$m_k = s_{m,k}M, \text{ where } s_{m,k} = \frac{s'_{m,k}}{\sum_j s'_{m,j}} \tag{20}$$

$$I_{0,k} = m_k L_k^2 \Lambda_k \tag{21}$$

Lastly, we compute the **body weight** $W$ of the human in Newtons, with $g = 9.8m/s^2$ as

$$W = g\sum_k m_k \tag{22}$$

### A.2  GRFM Model

Let $\mathcal{F} = [\mathbf{F}, \mathbf{M}]^{\mathsf{T}}$ be the ground reaction forces and moments (GRFM) applied at the CoM of a foot in global cartesian coordinates. Let $\mathbf{F} = [F_X, F_Y, F_Z]^{\mathsf{T}}$ where $Y$ is the vertical direction, and $\mathbf{z} = [z_x, z_y, z_z]^{\mathsf{T}}$ be the center of pressure (CoP) in the foot's local coordinates where $x$ is along the length of the foot, such that

$$\mathbf{M} = R_{ankle}^0 \mathbf{z} \times \mathbf{F} \tag{23}$$

where $R_k$ is joint $k$'s local rotation matrix, and $R_k^0 = R_{p(k)}^0 R_k$ describes the chain of rotational transformations from the world frame to its local frame. We use lower case $x, y, z$ to denote the foot's local coordinates, and its dimensions $\{l_l, l_w, l_h\}$ as shown in Fig. 6.

We predict the force in the vertical direction scaled by body weight $F_Y^W = F_Y/W$, and CoP along the foot scaled by foot length $z_x^l = z_x/l_l$, from initial estimates based on the foot's kinematics $\Psi$ and linear coefficients $\boldsymbol{\eta}$. Furthermore, let $\mu$ be the **coefficient of friction**, initialized at 0.8. With our regressed residuals $\delta$, and binary contact $\mathbf{c}$, we infer:

$$\{F_Y^W, z_x^l, \mu\} = \{\boldsymbol{\eta}_{FY}\Psi, \ \boldsymbol{\eta}_{zx}\Psi, \ 0.8\} + \delta_{\{Y,l,\mu\}} \tag{24}$$

$$F_Y = F_Y^W \cdot \text{body weight} \tag{25}$$

$$z_x = z_x^l \cdot l_l \tag{26}$$

Specifically, $\Psi = [1, P_{ankle,Y}, P_{oppAnkle,Y}, \dot{P}_{ankle,Y}, \ddot{P}_{ankle,Y}, q_{ankle,z}, \dot{q}_{ankle,z}, \ddot{q}_{ankle,z},]$ includes the ankle's linear kinematics in the direction opposite of gravity, and angular kinematics corresponding to plantar/dorsiflexion. Linear coefficients were fitted on the forceplate data in [72], with $\boldsymbol{\eta}_{FY} = [0.3116, 3.1785, -2.2963, 0.4151, 0.0088, 0.3374, -0.1206, -0.0089]$ and $\boldsymbol{\eta}_{zx} = [0.68996, -3.1508, 0.5925, 0.21997, 0.0035, 0.18502, -0.03311, -0.00212]$.

The remaining $\delta$ terms are scaling factors between -1 and 1 to ensure the values in the other directions are physically possible (*i.e.* $F_X^2 + F_Z^2 \leq \mu^2 F_Y^2$ and $z$ is within the foot's dimensions)

$$F_X = \delta_X \mu F_Y, \qquad\qquad F_Z = \delta_Z \sqrt{\mu^2 F_Y^2 - F_X^2} \qquad (27)$$

$$z_y = -|\delta_h l_h|, \qquad\qquad z_z = \delta_s(l_w/2) \qquad (28)$$

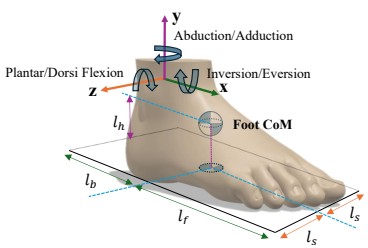

Figure 6: Foot local coordinate system and dimensions, with length $l_l = l_f + l_b$, width $l_w = 2l_s$, and CoM height $l_h$.

## A.3 Muscle torque generators

We compute MTG torque $\boldsymbol{\tau}_{MTG}$ using the equations below that are parameterized by the $\gamma$ coefficients that can be found in the tables of [54, 53]. For each joint rotational DoF $q \in \boldsymbol{q}_{[6:]}$, with angular velocity $\dot{q}$, let muscle signal $\alpha \in [0, 1]$ represent the joint's corresponding activation level for this DoF, we separate its $\tau_{MTG}$ into active torque generation $\tau_{active}$ and passive impedance $\tau_{passive}$

$$\tau_{MTG} = \tau_{active} + \tau_{passive} \qquad (29)$$

We compute the **active torque** as

$$\tau_{active} = \alpha \tau_\omega \tau_\theta \tau_0 \qquad (30)$$

where $\tau_\omega$ models the **active-torque–angular-speed relationship** [69, 67] and is paramterized as a piecewise function with coefficients $\gamma_{1:3}$.

$$\tau_\omega(\dot{q}) = \mathbb{1}_{\dot{q}<0}\left(\frac{(1-\gamma_1)|\omega_{max}| - (\gamma_2+1)\gamma_1\gamma_3\dot{q}}{(1-\gamma_1)|\omega_{max}| + (\gamma_2+1)\gamma_1\dot{q}}\right) + \mathbb{1}_{\dot{q}\geq0}\left(\frac{|\omega_{max}| - \dot{q}}{|\omega_{max}| + \gamma_2\dot{q}}\right) \qquad (31)$$

The peak velocity $\omega_{max}$ for each joint we use the values from [54, 53]. The coefficient $\gamma_1$ is the ratio of the maximum eccentric isokinetic torque over the maximum isometric torque [69, 15], $\gamma_2$ is the slope of the eccentric and concentric functions when the angular velocity is zero [69], and $\gamma_3$ is a shape factor that influences the curvature of the hyperbola in the torque-velocity concentric relationship [4].

$\tau_\theta$ models the **active-torque-angle relationship** [21, 33] and is represented by the non-negative portion of a polynomial (32) with coefficients $\gamma_{4:6}$

$$\tau_\theta(q) = (\gamma_4 + \gamma_5 q + \gamma_6 q^2)_+ \qquad (32)$$

$\tau_0$ is the **peak isokinetic torque** that controls peak MTG output at zero joint velocity, which can be measured via dynamometry.

$\tau_{passive}$ is the **passive torque** [1] of a joint that arises when the surrounding muscles, tendons, and ligaments are strained and intensifies near anatomical joint limits [1, 24, 81]. A joint's viscous damping and nonlinear stiffness are commonly described by a double exponential function [79]

$$\tau_{passive} = \gamma_{10e}^{-\gamma_{11}(\boldsymbol{q}-\boldsymbol{q}_{min})} - \gamma_{12}e^{\gamma_{13}(\boldsymbol{q}-\boldsymbol{q}_{max})} - \gamma_{14}\omega \qquad (33)$$

where $\gamma_{10-14}$ are passive coefficients from [48] and $\gamma_{11}$ is the rotational damping linear coefficient [78] to reflect viscoelasticity. This encourages the joint to move within its **range of motion** (RoM), as a large restoring torque is produced otherwise.

## A.4 Inverse dynamics [1]

We compute $\boldsymbol{\tau}_q$ using Lagrange's equations derived from d'Alembert's Principle of virtual work

$$\boldsymbol{\tau}_q = \mathfrak{M}\ddot{\boldsymbol{q}} + \mathfrak{C} - \mathfrak{F} \tag{34}$$

We can write the terms on the right hand side as:

$$\mathfrak{M} = \sum_k J_k^\mathsf{T} \mathcal{M}_k J_k, \tag{35}$$

$$\mathfrak{C} = \sum_k (J_k^\mathsf{T} \mathcal{M}_k \dot{J}_k + J_k^\mathsf{T} \begin{bmatrix} 0 & 0 \\ 0 & [J_{\Omega,k}\dot{q}]_s \end{bmatrix} \mathcal{M}_k J_k)\dot{\boldsymbol{q}} \tag{36}$$

$$\mathfrak{F} = J_{LFoot}^\mathsf{T} \mathcal{F}_{LFoot} + J_{RFoot}^\mathsf{T} \mathcal{F}_{RFoot} \tag{37}$$

where $\mathbf{I}_3$ is the identity matrix, $([\cdot]_s)$ denotes the skew-symmetric form, and

$$\mathcal{M}_k = \begin{bmatrix} m_k \mathbf{I}_3 & 0 \\ 0 & R_k^0 I_{0,k}(R_k^0)^\mathsf{T} \end{bmatrix} \tag{38}$$

To deal with potential energy, we offset the root acceleration in the direction of gravity by -9.8 m/s$^2$. Jacobian matrix $J$ is the mapping from the generalized space to the global Cartesian coordinates, such that for linear and angular velocities $\boldsymbol{V}_k, \boldsymbol{\Omega}_k$ in global Cartesian coordinates, we have:

$$J_k \dot{\boldsymbol{q}} = \begin{bmatrix} J_{V,k} \\ J_{\Omega,k} \end{bmatrix} \dot{\boldsymbol{q}} = \begin{bmatrix} \boldsymbol{V}_k \\ \boldsymbol{\Omega}_k \end{bmatrix} \tag{39}$$

$J$ can be computed analytically using a recursive algorithm such as in [17]. For segment $k$, we define its parent segment $p(k)$ as its neighboring segment that is closer to the root. Other than the root, each segment has one and only one parent. We define $k$'s children $ch(k)$ as its neighboring segments further away from the root. Let $\boldsymbol{r}_{a\to b}$ denote the 3D displacement from point $a$ to $b$. For segment $k$, with linear velocity $\boldsymbol{V}_k$ at its CoM and linear velocity $\boldsymbol{V}_k^{joint}$ at its corresponding joint, we have

$$\boldsymbol{V}_k^{joint} = \boldsymbol{V}_{p(k)} + \boldsymbol{\Omega}_{p(k)} \times \boldsymbol{r}_{p(k)\to k^{joint}} \Rightarrow J_{V,k}^{joint} = J_{V,p(k)} - [\boldsymbol{r}_{p(k)\to k^{joint}}]J_{\Omega,p(k)} \tag{40}$$

and the velocity at the CoM of segment $k$ becomes:

$$\boldsymbol{V}_k = \boldsymbol{V}_k^{joint} + \boldsymbol{\Omega}_k \times \boldsymbol{r}_{k^{joint}\to k} \Rightarrow J_{V,k} = J_{V,p(k)} - [\boldsymbol{r}_{p(k)\to k^{joint}}]J_{\Omega,p(k)} - [\boldsymbol{r}_{k^{joint}\to k}]J_{\Omega,k} \tag{41}$$

From (41), we can compute the time derivative recursively as:

$$\dot{J}_{V,k} = \dot{J}_{V,p(k)} - [\boldsymbol{r}_{p(k)\to k^{joint}}]\dot{J}_{\Omega,p(k)} - [\boldsymbol{r}_{k^{joint}\to k}]\dot{J}_{\Omega,k} \tag{42}$$

The global angular velocity of $k$ in skew symmetric form is:

$$[\boldsymbol{\Omega}_k] = \dot{R}_k^0 (R_k^0)^\mathsf{T} = (R_{p(k)}^0 R_k)^{\dot{}}(R_{p(k)}^0 R_k)^\mathsf{T} \tag{43}$$

$$= ... = \dot{R}_{p(k)}^0 (R_{p(k)}^0)^\mathsf{T} + R_{p(k)}^0 (\dot{R}_k R_k^\mathsf{T})(R_{p(k)}^0)^\mathsf{T} \tag{44}$$

$$= [\boldsymbol{\Omega}_{p(k)}] + R_{p(k)}^0 [\boldsymbol{\omega}_k](R_{p(k)}^0)^\mathsf{T} \qquad (\because [\boldsymbol{\omega}_k] = \dot{R}_k R_k^\mathsf{T}) \tag{45}$$

$$\Rightarrow \boldsymbol{\Omega}_k = \boldsymbol{\Omega}_{p(k)} + R_{p(k)}^0 \boldsymbol{\omega}_k \qquad (\because [A\boldsymbol{b}] = A[\boldsymbol{b}]A^\mathsf{T}) \tag{46}$$

To avoid confusion of notation, we also write joint $k$'s rotation $\boldsymbol{\theta}_k \triangleq \boldsymbol{q}_k$, i.e. we have generalized coordinates $\boldsymbol{q} = \begin{bmatrix} \boldsymbol{X}_0 \\ \boldsymbol{\theta} \end{bmatrix} \in \mathbb{R}^{N_{DoF}\times 1}$ where $\boldsymbol{X}_0$ is the global root translation, $\boldsymbol{\theta}_0$ is the global root rotation, $\boldsymbol{\theta}_k$ describes the local rotation of segment $k$ relative to its parent $p(k)$, and $\boldsymbol{\theta}^\mathsf{T} = [\boldsymbol{\theta}_0^\mathsf{T} \ \boldsymbol{\theta}_1^\mathsf{T} \ ... \ \boldsymbol{\theta}_{N_k}^\mathsf{T}]$. Let $J_{\omega,k}$ be the local Jacobian such that $\boldsymbol{\omega}_k = J_{\omega,k}\dot{\boldsymbol{\theta}}_k$. We can compute $\boldsymbol{\Omega}_k$ recursively:

$$\boldsymbol{\Omega}_k = \boldsymbol{\Omega}_{p(k)} + R_{p(k)}^0 J_{\omega,k}\dot{\boldsymbol{\theta}}_k \tag{47}$$

$$= 0 + J_{\omega,0}\dot{\boldsymbol{\theta}}_0 + ... + R_{p(p(k))}^0 J_{\omega,p(k)}\dot{\boldsymbol{\theta}}_{p(k)} + R_{p(k)}^0 J_{\omega,k}\dot{\boldsymbol{\theta}}_k \tag{48}$$

$$\triangleq J_{\Omega,k}\dot{\boldsymbol{q}} \tag{49}$$

---

[1] Derivations in this section are based on C. Karen Liu and Sumit Jain's multibody dynamics notes: `https://fab.cba.mit.edu/classes/865.18/design/optimization/dynamics_1.pdf`.

Let $\mathcal{P}_k$ denote the set of all ancestors of $k$ and itself ($k \in \mathcal{P}_k$), we split $J_{\Omega,k}$ into $N_k + 1$ blocks of size $3 \times 3$:

$$J_{\Omega,k} = \begin{bmatrix} 0_{3\times3} & J_{\omega,0} & \mathbb{1}_{1\in\mathcal{P}_k} R_{p(1)}^0 J_{\omega,1} & ... & \mathbb{1}_{N_k\in\mathcal{P}_k} R_{p(K)}^0 J_{\omega,N_k} \end{bmatrix} \in \mathbb{R}^{3\times N_{DoF}} \qquad (50)$$

For segment $k$, if we represent rotation $\boldsymbol{\theta}_k = \begin{bmatrix} \theta_{k,1} & \theta_{k,2} & \theta_{k,3} \end{bmatrix} \triangleq \begin{bmatrix} \alpha & \beta & \gamma \end{bmatrix} \in \mathbb{R}^3$ using 3 Euler angles, removing subscript $k$ for notation simplicity, we have

$$[\boldsymbol{\omega}] = \dot{R}R^\mathsf{T} = \sum_i \frac{\partial R}{\partial \theta_i} R^\mathsf{T} \dot{\theta}_i = \frac{\partial R}{\partial \alpha} R^\mathsf{T} \dot{\alpha} + \frac{\partial R}{\partial \beta} R^\mathsf{T} \dot{\beta} + \frac{\partial R}{\partial \gamma} R^\mathsf{T} \dot{\gamma} \qquad (51)$$

and it remains to compute $\mathbf{J}$'s s.t.

$$J_\omega \triangleq \begin{bmatrix} \mathbf{J}_1 & \mathbf{J}_2 & \mathbf{J}_3 \end{bmatrix}, \quad \text{where } [\mathbf{J}_1] = \frac{\partial R}{\partial \alpha} R^\mathsf{T}, \quad [\mathbf{J}_2] = \frac{\partial R}{\partial \beta} R^\mathsf{T}, \quad [\mathbf{J}_3] = \frac{\partial R}{\partial \gamma} R^\mathsf{T} \qquad (52)$$

which satisfies $[\boldsymbol{\omega}] = \sum_i [\mathbf{J}_i]\dot{\theta}_i \quad \text{and} \quad \boldsymbol{\omega} = J_\omega \dot{\boldsymbol{\theta}} \qquad (53)$

Finally, let segment $l \in \mathcal{P}_k$ be an ancestor of $k$ and denote $\{\boldsymbol{J}_{\Omega,k}\}_l \triangleq R_{p(l)}^0 J_{\omega,l}$ as the $(l+2)$-th $3 \times 3$ block in $J_{\Omega,k}$ from (50), we can compute its time derivative as:

$$\{\dot{\boldsymbol{J}}_{\Omega,k}\}_l = \dot{R}_{p(l)}^0 J_{\omega,l} + R_{p(l)}^0 \dot{J}_{\omega,l}, \quad \text{with } \dot{J}_{\omega,l} = \sum_{l,i} \frac{\partial J_{\omega,l}}{\partial \theta_{l,i}} \dot{\theta}_{l,i} \qquad (54)$$

following (52), it remains to compute $\dot{\mathbf{J}}$'s s.t.

$$\dot{J}_{\omega,l} \triangleq \begin{bmatrix} \dot{\mathbf{J}}_{l,1} & \dot{\mathbf{J}}_{l,2} & \dot{\mathbf{J}}_{l,3} \end{bmatrix}, \quad \text{where } \dot{\mathbf{J}}_{l,j} = \sum_{l,i} \frac{\partial \mathbf{J}_{l,j}}{\partial \theta_{l,i}} \dot{\theta}_{l,i} \qquad (55)$$

## A.5 Neural network

We trained a transformer encoder consisting of 8 layers with a latent dimension of 256, using total loss $\mathcal{L}_{total}$ with weights $\boldsymbol{\lambda}_{kin} = [0.5, 10, 1000, 1, 20, 1000]$ and $\boldsymbol{\lambda}_{dyn} = [100, 100, 20]$. We trained on AMASS with a sequence input length of 16 frames, after removing sequences containing non feet-ground contact, with contact labels from [83], for 25 epochs. We used the AdamW optimizer [44] with a weight decay of $10^{-4}$ and an initial learning rate of $10^{-4}$ that decreases by 20% every 5 epochs. The entire process can be trained in about 12 hours on a single Titan Xp GPU.

