# OpenReview forum: "3D Human Pose Estimation with Muscles"
_NeurIPS.cc/2025/Conference — NeurIPS 2025 poster_

### Official Review · Reviewer_TsEp · 2025-06-17

**Clarity:** 3
**Significance:** 3
**Originality:** 3
**Rating:** 4
**Confidence:** 3

**Summary:**

The paper introduces MusclePose, an end-to-end learnable physics-infused 3D human pose estimator that integrates muscle dynamics modeling to infer human dynamics from monocular videos. Unlike existing physics-based pose estimators that rely on external physics optimizers or produce implausible kinetics, MusclePose employs a multihypothesis approach to compute joint torques via Lagrange’s equations and muscle torque generators (MTGs). It also outputs detailed human anthropometrics based on biomechanics studies, enhancing biofidelity.

**Questions:**

As shown in the Weaknesses section, A reasonable explanation will make me raise the score for this work.

**Ethical Concerns:**

["NO or VERY MINOR ethics concerns only"]

**Final Justification:**

This is a work with a solid foundation in theory. Considering the comments from other reviewers and the author's responses, I maintain my score.

**Limitations:**

Yes, limitations are discussed in the supplementary material.

**Paper Formatting Concerns:**

1. line 82: "are different" -> "is different"

**Quality:**

3

**Strengths And Weaknesses:**

## Strengths

1. Compared with PhysPT, the proposed ground reaction forces and moments (GRFM) have better mathematical modeling.
2. The introduced MTGs (Muscle Torque Generators) can generate more reasonable joint torques while providing the muscle activation levels.

## Weaknesses
### Major
1. PhysPT said that "The discrete contact status also introduces a non-differentiable process in estimating the forces." How does GRFM deal with this issue?

### Minor
1. Does the MTG model all joints uniformly? Is it currently less effective for joints with 3 DOF (e.g., shoulders)? How does the paper ensure that fitting errors in complex joints do not propagate to the global dynamics estimation?
2. Could you clarify the specific location of MTG in Reference [46] (cited in lines 63 and 207)? A more detailed explanation of the MTG implementation would be helpful.
3. Why are Coriolis and centrifugal forces included in the Lagrangian formulation? Is the algorithm's reference frame the global/world coordinate system?

---

> ### Author Rebuttal · Authors · 2025-07-30
>
> We thank you for your thoughtful feedback. We are encouraged you found our method impactful in producing better kinetics estimates. We have responded to the your Weaknesses ($\textbf{W}$) section below, and will incorporate them into our submission. We hope our explanations can improve the clarity of the paper.
>
> $\textbf{W1, on ground reaction forces and moments (GRFM) estimation:}$ we followed the same approach as PhysPT (and other regression-based physics pose estimators), and estimated GRFM based on the foot's mesh height to the ground, and other foot kinematics, which are all non-discrete. One main difference is that we made use of existing biomechanics datasets [66] to initialize our GRFM model parameters (L663-667 of the Supp).
>
> $\textbf{W2(a), does the MTG model all joints uniformly?}$
> No.
> While all DOFs are modeled using Eq 31-35 in the Supp, each DOF has its own set of MTG coefficients $\gamma$'s derived from different biomechanics studies (values summarized in Tabs. 4,5 of [48] and Tabs 4,5. of [49]).
> This joint-dependent parameterization preserves physiological realism (e.g., hip flexion and knee extension exhibit different peak torques and passive stiffness profiles), unlike uniform torque models that assume identical properties across the body.
>
> $\textbf{W2(b), are MTGs less effective for complex joints?}$
> Possibly.
> Existing evaluations of complex joint such as the shoulders are limited [46]. As a result, we allowed the maximum isokinetic torque and range of motion of all DOFs to be learnable, instead of being fixed or scaled by human dimensions, as in biomechanics studies [48,49].
>
> $\textbf{W2(c), limiting effects of errors in complex joints on global dynamics estimation:}$
> these errors are more pronounced in distal segments (the diagonal of their inverted mass matrices are larger), hence we moved the DOF for forearm pronation/supination from the wrist to the elbow. DOFs for the ankles are constrained from the GRFM model, as errors can result in unrealistic forces. Lastly, redundant DOFs were removed, such as the one in the scapula that coincides with shoulder flexion/extension.
>
> $\textbf{W3, location of MTG in [46] (cited in L63,207):}$
> Eq. 4 in [46], which we expanded and rewrote as Eq 31-35 in App. B.4 of the Supp.
> We used the set of MTG coefficients in Tabs. 4,5 of [48] and Tabs. 4,5 of [49], and allowed the maximum isokinetic torque and range of motion of all DOFs to be learnable.
> We will expand on the MTG details in App. B.4 of the Supp.
> A more complete explanation of MTGs can be found in [Inkol et al, 2020, Muscle torque generators in multibody dynamic simulations of optimal sports performance].
>
> $\textbf{W4(a), why are Coriolis and centrifugal forces included:}$
> these are components of the equations of motion derived via Lagrange’s method (Eq. 1, 11). These second-order effects become significant when angular velocities are high or when multi-joint interactions occur, such as in baseball pitching or golf swings, where rapid rotations of the torso and shoulders generate substantial inertial coupling effects. Neglecting these terms would lead to systematically biased torque estimates, particularly for proximal joints where interaction forces dominate.
>
> $\textbf{W4(b), is the algorithm's reference frame the global/world coordinate system?}$
> Yes. Specifically, the root is defined in world coordinates (we used the canonical form in [77], and centered the non-gravity axes at 0 at the start of each input sequence), while joint rotations are represented in the local coordinates consistent with the International Society of Biomechanics (ISB) format.
> This hybrid representation allows us to maintain physical consistency (e.g., direction of gravity) to better compute ground contact, Coriolis force, etc., while the latter enables accurate relative joint modeling and compatibility with MTG models.

---

> > ### Comment · Reviewer_TsEp · 2025-08-06
> >
> > The author's reply addressed most of my concerns, but there is still a small issue: whether the MTG uses an absolute 3D coordinate system, which excludes the Coriolis force and centrifugal force.

---

> ### Author Response · Authors · 2025-08-07
>
> No, MTGs use local joint coordinate systems.
>
> $\textbf{Generalized force $Q$}$.
> Let $N_k$ be the total number of joints, $N_{DOF}$ be the total number of DOFs, and $q=[X_0, q_0, q_1, ...,q_{N_k}] \in \mathbb{R}^{N_{DOF}}$ be the $\textit{generalized coordinates}$ in which we want to express motion in, to model Lagrangian dynamics. Here $X_0$ and $q_0$ correspond to the root's translational and rotational DOFs in global coordinates, and the remaining $q_k$'s each describe a joint's rotational DOFs in the joint's local coordinate system.
> We denote $Q\in\mathbb{R}^{N_{DOF}}$ as the $\textit{\textbf{generalized force}}$ in this generalized space,
> where each entry of $Q$ represents the total force applied on each DOF in its corresponding joint's local coordinate system.
> This also makes evaluations and comparisons easier, as the magnitudes of the non-root entries of $Q$ won't depend on an arbitrarily set global cartesian system.
>
> $\textbf{Coriolis and centrifugal forces}$
> are not additional "physical forces" to model, and are just names we assigned to the components of a "bias" term ($C$ in Eq. R1 below or Eq. 38 in the Supp) that appears in Lagrange's equations, which express dynamics in the generalized space.
>
> (rewriting Eq. 1 with simplified notations to better fit the response format)
> $$
> \underset{\substack{\text{generalized} \\\ \text{force}}}{Q} =
> \underset{\substack{\text{"mass" times} \\\ \text{accel term}}}{M\ddot{q}} +
> \underset{\substack{\text{"bias"} \\\ \text{term}}}{C} =
> \underset{\text{Jacobian}}{J^\top} \cdot
> \underset{\substack{\text{cartesian} \\\ \text{force\\&moments}}}{F}
> \qquad\qquad\text{(R1)}
> $$
> Intuitively, from Eq. R1, we can think of $C$ as a bias term that arises when we project dynamics from the cartesian space to the generalized space (i.e. from $F$ to $Q$), since the former's coordinate system is fixed whereas the latter's is expressed in local coordinates relative to each joint's parent, which moves.
> This $C$ term is "embedded" into the Jacobian matrix $J$ (computed in Eq. 41-57 of the Supp), which handles this conversion.
>
> $\textbf{Joint torques }$ $\tau_q=Q - J^\top F_{ext}$ are isolated from $Q$ by separating the effects of the external forces $J^\top F_{ext}$ (e.g. GRFMs) in the generalized space.
>
> $\textbf{On using MTGs to estimate $\tau_q$ (with local joint coordinates):}$ as described earlier, since $\tau_q\in\mathbb{R}^{N_{DOF}}$ is in the generalized space, the non-root entries of $\tau_q$ are expressed in local joint coordinates, which is what MTGs use.
> The entries in $\tau_q$ that correspond to the root DOFs (global cartesian coordinates) are assumed to be zero.
> From Eq. R1, we can convert $\tau_q$ to cartesian coordinates $F_q = (J^\top J)^{-1}J \tau_q$ using the pseudo inverse of $J^\top$, which handles the $C$ term.
>
> ---
> ---
>
> ***Some details on conversion from $F$ to $Q$:***
>
> From the Newton-Euler equations, we can compute cartesian forces $f$ and moments $\mathcal{T}$ from cartesian linear velocities $V$ and angular velocities $\Omega$ as
> \begin{align}
>     F =
>     \begin{bmatrix}
>         {f} \\\\
>         {\mathcal{T}}
>     \end{bmatrix}
>     &=
>     \begin{bmatrix}
>         m \mathbf{I}_3 & 0 \\\\
>         0 & I_c
>     \end{bmatrix}
>     \begin{bmatrix}
>         {\dot{V}} \\\\
>         {\dot{\Omega}}
>     \end{bmatrix}
>     +
>     \begin{bmatrix}
>         0 \\\\
>         {\Omega} \times I_c\Omega
>     \end{bmatrix}
> \end{align}
> with segment masses $m$, inertia tensors $I_c$, and identity matrix $\mathbf{I}_3$.
>
> Substituting in generalized coordinates $q$ using $J= \begin{bmatrix} J_V \\\\ J_\Omega \end{bmatrix}$ , and denoting $M_c= \begin{bmatrix} m \mathbf{I}_3 & 0 \\\\ 0 & I_c  \end{bmatrix}$ and ($[\cdot]_s$) as the skew-symmetric form of a vector, we have
>
>
> \begin{align}
>     F
>     &= M_{c} (\dot{J \dot{q}})
>     +
>     \begin{bmatrix}
>         0 \\\\
>         (J_\Omega \dot{q}) \times I_c J_\Omega\dot{q}
>     \end{bmatrix}
>     \\\\
>     &= M_{c} J {\ddot{q}}
>     + (M_{c} \dot{J} +  [J_\Omega \dot{q}]_s M_c J) \dot{q}
> \end{align}
>
> Multiplying both sides by $J^\top$ give us Lagrange's equations (in generalized space):
>
> \begin{align}
>     J^\top F
>     =
>     J^\top M_c J \ddot{q}
>     +
>     (J^\top M_c \dot{J} +  J^\top[J_\Omega \dot{q}]_s M_c J) \dot{q}
>     =
>     Q
> \end{align}
>
> where the second term $C=(J^\top M_c \dot{J} +  J^\top[J_\Omega \dot{q}]_s M_c J) \dot{q}$ is the aforementioned "bias" term.
>
> (for more details, see App B.5 of the Supp)

---

### Official Review · Reviewer_xJ9Z · 2025-06-30

**Clarity:** 2
**Significance:** 2
**Originality:** 2
**Rating:** 4
**Confidence:** 3

**Summary:**

This paper introduces MusclePose, a 3D human pose estimator that integrates muscle-dynamics modeling and detailed anthropometrics for inferring human dynamics from monocular video. The authors claim that MusclePose achieves competitive positional accuracy while providing more physically plausible kinetic inferences compared to existing methods.

**Questions:**

(1) The "kinematics-kinetics trade-off" is mentioned as a possible reason for slightly worse positional accuracy on 3DPWoc. Could you elaborate on this trade-off in the context of MusclePose? How does the model balance enforcing physical plausibility (kinetics) with maintaining high positional accuracy (kinematics)? Is there a tuning mechanism or a loss weighting that allows for prioritizing one over the other, and if so, how was it determined?

(2) The paper lacks qualitative results showcasing MusclePose's performance on real-world images or videos. Could the authors provide visual examples of predicted 3D poses and body shapes overlaid on input video frames to demonstrate the model's output in a real-world context?

(3) PhysPT, a competing method, employs several physical plausibility metrics (ACCL, VEL, FS, GP) and evaluates on "all actions". In contrast, MusclePose's evaluation is limited to sequences with "feet-ground contact only". Could the authors justify the omission of a broader evaluation set and these standard physical plausibility metrics, and explain how the presented results still provide a fair comparison with state-of-the-art methods like PhysPT?


(4) The "with \alpha " case in the ablations initializes muscle signals based on angular velocity relative to its limit, with an additional offset term regressed. Could you provide more details on why this specific initialization was chosen and what impact it has on the learning process and the resulting muscle signals, especially considering the observed discrepancies in Figure 4 (e.g., for ankle plantarflexion)?

**Ethical Concerns:**

["NO or VERY MINOR ethics concerns only"]

**Final Justification:**

All of my concerns are resolved during the rebuttal, therefore I raise my score from 3 to 4.

**Limitations:**

Yes

**Paper Formatting Concerns:**

There is no major concern

**Quality:**

2

**Strengths And Weaknesses:**

Strengths:

(1) Novelty in Physics-Infused HPE: The core strength of MusclePose lies in its novel integration of muscle-dynamics modeling and detailed anthropometrics directly into an end-to-end learnable physics-infused 3D human pose estimator. This goes beyond typical physics-based pose estimation methods that often rely on external physics optimizers or treat kinetics as auxiliary predictions.

(2) Improved Biofidelity of Kinetic Inference: The paper demonstrates that MusclePose can infer plausible human kinetics and muscle signals consistent with biomechanics studies, without requiring an external physics engine. This is an improvement over prior regression-based approaches, where kinetic predictions may not be physically plausible. The use of a multihypothesis approach, inferring joint torques via Lagrange's equations and muscle dynamics modeling, contributes to this biofidelity.

(3) Detailed Anthropometric Prediction: MusclePose's ability to predict detailed human anthropometrics based on biomechanics studies is an advancement, contrasting with existing methods that use simpler shape primitives. The ablations confirm that this custom anthropometric prediction leads to better kinetics.

Weaknesses:

(1) Limited Scope of External Force Modeling and the Evaluation Setting: The paper assumes only foot-ground contact for simplicity. While this simplifies the problem and the formulation, it limits the applicability of MusclePose to scenarios involving more complex interactions with the environment or multiple external forces, which are acknowledged as underconstrained problems. Moreover, the paper states that it "trained and evaluated on sequences with feet-ground contact only" and explicitly removed non-feet-ground contact sequences from both H36M and 3DPWoc datasets for evaluation. This narrower evaluation scope contrasts directly with PhysPT, which claims to "adhere to the standard protocol and evaluate our method on all actions," implying evaluation on sequences beyond just ground contact. This discrepancy in evaluation protocols makes direct comparisons of generalizability and robustness between MusclePose and methods like PhysPT unfair, as MusclePose's performance on more diverse actions (e.g., sitting, climbing, freestyle movements) remains unassessed.

(2) Qualitative Evaluation of Muscle Signals: While the paper states that MusclePose can infer plausible muscle signals, the evaluation of these signals is primarily qualitative, comparing them to trends from biomechanics studies and EMG data. A more rigorous quantitative assessment, if possible, would strengthen the claim.

(3) Limited Evaluation Metrics: While the paper focuses on MJE/PJE for positional accuracy, it's true that PhysPT, a competing method, utilizes additional physical plausibility metrics such as ACCL (acceleration), VEL (velocity), FS (foot sliding), and GP (ground penetration). The omission of these quantitative physical plausibility metrics in the main evaluation for MusclePose is a notable limitation. This limits a direct and comprehensive comparison of the physical plausibility aspects between MusclePose and PhysPT, especially since MusclePose's core contribution is enhanced biofidelity in kinetics. While the paper includes quantitative metrics like residual forces and out-of-range joint torques (Table 2) , and qualitative comparisons of joint torques and muscle signals against biomechanics studies (Figure 3 and 4), the inclusion of standard physical plausibility metrics like ACCL, VEL, FS, and GP would provide a more complete picture of its physical realism from a kinematic perspective.

(4) Under-discussed Computational Efficiency: While the paper claims inference speed on par with purely kinematic pose estimators, a more detailed analysis of the computational cost and real-time performance of MusclePose compared to its baselines would be beneficial.

---

> ### Author Rebuttal · Authors · 2025-07-31
>
> We thank you for your thoughtful feedback. We are encouraged that you found our method novel and an improvement over existing regression-based physics human pose estimators (PHPE). We hope that our responses to Weakness $\textbf{W1-3}$ below can improve the significance of our paper, and that our answers to $\textbf{W4}$ and questions $\textbf{Q1-4}$ resolves the clarity concerns. If you have additional suggestions on how we can improve on overall paper quality and originality, please let us know.
>
> $\textbf{W1(a), limited scope of external force modeling:}$
> we, along with most regression-based PHPE [33,55,57,71], assume the feet-ground contact only scenario due to a lack of large-scale video datasets with contact labels, unlike some optimization-based methods [17, 75] that use a physics simulator. Under this scenario, when both feet are in contact with the ground, the dynamics problem is not simplified as it remains underconstrained. As discussed in L613-615 of the Supp, without loss of generality, we can expand our method to be trained and evaluated on datasets with more contact labels by simply adding contact points to our human model. While PhysPT includes additional contact points, by the same sense it remains a "simplified" case, as the contact points don't cover the entire body of the human. Furthermore, as shown in Fig. 7 of PhysPT, feet-ground contact takes up the significant majority of the contact events in their training data.
>
> $\textbf{W1(b), on the narrower evaluation scope leading to unfair comparisons with PhysPT:}$
> we respectfully disagree with this comment. Comparisons on H36M (Tab. 3, Figs. 3,5), 3DPWoc (Tab. 4),  PennAction (Fig. 3), were done with the same data and procedure (after removal of non feet-ground contact).
>
> $\textbf{W1(c), on limited implications due to the narrower evaluation scope:}$
> again, we respectfully disagree.
> The "standard protocol" mentioned in PhysPT refers to the evaluation procedure for purely kinematic pose estimators, while regression-based PHPE often evaluates on the feet-ground contact only scenario, described in \textbf{W1(a)}.
> While certain sequences (e.g. sitting) are removed in the latter case, the majority (77\%) of the data remains, and contains poses that are kinematically similar (e.g. squatting) to the removed ones.
> What remains unevaluated is the estimated kinetics of these removed scenarios, which we do not claim to model, and which PhysPT also did not report.
> However, for the feet-ground contact case, which is the majority, we showed improvements in kinetics over PhysPT.
>
>
> $\textbf{W2, lack of rigorous muscle assessment:}$
> our current muscle evaluation in Fig. 4 follows the procedure in [25], to show consistencies in the timings and trends of our muscle activation predictions with biomechanics studies, as we are not aware of public video datasets with corresponding muscle activation levels to directly compare with.
> While we do compare with EMG data, the predicted muscles signals are more of a surrogate representation of EMGs, and the two are not exactly the same. Hence, mismatches in magnitude will exist. Furthermore, raw EMG values can vary widely due to electrode placement.
>
> $\textbf{W3, lack of kinematic plausibility metrics:}$
> these metrics are not omitted and are reported in Tabs. 3,4 of the submitted Supp, which also includes additional metrics to further investigate GP, by evaluating the inferred ground reaction forces as an alternative, and by evaluating its opposite case -- floating. We showed that our method had similar kinematic plausibility as PhysPT, both of which improves over purely-kinematic methods, and that our method further improves on kinetic plausibility when compared to PhysPT.
>
>
> $\textbf{W4, on computational efficiency:}$
> as with most regression-based PHPE, inference time is largely based on the existing kinematics estimator used to generate initial estimates, unlike methods that require an external physics engine for optimization, which are often too slow for real-time inference (Tab. 1 of [17]).
> Specifically, our method adds about 0.003 seconds to the kinematics estimator used, for a video sequence of 16 frames, on a Titan Xp GPU.
>
>
> $\textbf{Q1, ``kinematics-kinetics trade-off" in 3DPWoc, and tuning:}$
> since this dataset was captured with a moving camera, the root degrees of freedom are harder to estimate, potentially leading to higher residual forces, which the model may try to reduce (lower kinetic error) by estimating a set of local joint kinematics slightly different from the original motion (higher kinematic error). \textbf{Tuning mechanism:} during training, the weights for the kinematic loss can be increased until the residual forces remain within a desired range.
>
> $\textbf{Q2, lack of qualitative results:}$
> these are not omitted and are shown in the top row of Fig 3. They were not overlayed on the images to better visualize the joint torques, align with the figures below, and to avoid cluttering, but we can add the images in the Supp.
>
> $\textbf{Q3:}$
> this is related to $\textbf{W1, W3}$. The metrics are not omitted as they are reported in Tabs. 3,4 of the Supp, and we believe that the comparison is fair as both methods were evaluated using the same procedure and data. Please see our responses for $\textbf{W1, W3}$.
>
>
> $\textbf{Q4(a), } \hat\alpha\textbf{ ablation:}$
> we added this ablation as an additional way to compare with existing methods that estimate joint torques as some linear combination of the joint's kinematics $\mathcal{K} = \\{ \text{joint angles } q, \text{angular velocity } \dot{q},... \\}$,
>
> $
> \text{ours: }\\; \tau_{MTG} = \alpha \cdot \tau_{active}(q, \dot{q}) + \tau_{passive}(q)
> $
>
> $
> \text{existing: }\\; \tau = b_0 + b_1 \dot{q} + b_2 q + b_3 F(\mathcal{K}),\qquad
> $
> (for learned coefficients $b$'s, and some parametric $F$)
>
> Since the muscle signal $\alpha$ can be interpreted as an ``effort level" between 0 and 1, for the ablation we estimated it as $\hat{\alpha} = (\dot{q} /\dot{q}_{limit}) + \epsilon$,
>
> where $\dot{q}_{limit}$ is the joint's anatomical angular velocity limit, and $\epsilon$ is a predicted offset term.
>
> $\textbf{Impact on learning:}$ the worsened results from this initialization could be due to its over-reliance on the initial kinematics estimates.
>
>
> $\textbf{Q4(b), muscle signal discrepancies in Fig.4:}$ as discussed in $\textbf{W2}$, while we see discrepancies in magnitudes between our predicted muscle signals and the EMG values, the two are not the same, and the comparison was made to show consistencies in the timings and trends of our muscle activations. Furthermore, for ankle plantarflexion, the magnitude discrepancy may also be due to differences in shear force generation, as the subjects in H36M walk in a small circle with slower and varying speeds, whereas the subjects in the biomechanics studies walk in a straight line at a consistent pace.

---

> > ### Comment · Reviewer_xJ9Z · 2025-08-05
> >
> > Dear authors,
> >
> > I sincerely appreciate the authors for addressing my concerns in the review, such as the limitation of using the external forces, the computational efficiency, and additional ablation studies.
> > My concerns are resolved by rebuttals.
> >
> > I hope that the comments are helpful to improve the quality of the paper.

---

### Official Review · Reviewer_m1ia · 2025-07-03

**Clarity:** 3
**Significance:** 3
**Originality:** 3
**Rating:** 5
**Confidence:** 4

**Summary:**

This paper proposed a learning framework for physics-based human pose estimation (PHPE) from monocular videos, taking into consideration of the kinetic constraints. The estimator accounted for individualized anthropometrics, showed improved accuracy compared to purely kinematics-based models, and demonstrated some degree of bio-fidelity.

**Questions:**

How accurate or realistic are the estimations of ground reaction forces and external forces, since the contact is simplified?
What's the inference efficiency?

**Ethical Concerns:**

["NO or VERY MINOR ethics concerns only"]

**Final Justification:**

I think the authors addressed most of the concerns by reviewers. So I maintain my recommendation for acceptance.

**Limitations:**

The authors discussed some limitations in the supplementary.

**Paper Formatting Concerns:**

No. But some figures are too small.

**Quality:**

3

**Strengths And Weaknesses:**

The proposed framework incorporated kinetics when estimating joint positions and individualized anthropometrics aiming to improve the pose estimation accuracy.  The estimator didn't need a physics engine nor musculoskeletal simulation and could be trained end-to-end, which makes it a potentially useful tool in computer vision. The model design rationale, architecture and implementations are clearly presented (with more details in supplementary). Evaluations were conducted on several standard 3D human pose estimation benchmarks, showing improved or comparable estimation accuracy, and some degree of bio-fidelity.

The improvements in pose estimation were not that significant. A MTG module was used for muscle activation-net joint torque estimation, which overlooked the detailed muscle dynamics and the redundancy problem was somehow avoided. Mismatches exist for the estimated human body and biomechanical studies. To better validate the bio-fidelity, simultaneous human behavioral and EMG recordings could be used. The motion included are mainly gross movements, further improvement may demonstrate the performance on fine-motor movements.

---

> ### Author Rebuttal · Authors · 2025-07-31
>
> We thank you for your thoughtful feedback. We are encouraged you found our method impactful and our paper clearly written. We have responded to your Weakness (W) section and questions (Q) below, and will incorporate them into our submission.
>
> $\textbf{W1, on modest kinematic improvements:}$ while this is true, the main focus and contribution of our work were to improve the biofidelity of kinetics estimations for regression based physics pose estimators, which we showed in Sec. 4.3 and App. A of the Supp.
>
> $\textbf{W2, on simplified muscle modeling:}$
> we used MTGs over complex muscle models to bypass the computational overhead of estimating detailed musculoskeletal geometries and redundant joint contact forces (both of which we don't need), while maintaining biofidelity (L64-74 and L139-145).
>
> $\textbf{W3, on mismatches with biomechanics studies:}$
> these additional evaluations were done to qualitatively assess the plausibility of estimated joint torques for certain well-studied human motions, in terms of trends and limits, when ground truth data is unavailable. Since the torques in these biomechanics studies are derived from different data and models, mismatches will exist.
>
> $\textbf{W4, on comparisons with EMGs:}$
> our muscle signals are compared to EMG data in Fig. 4.
> Note, the muscles signals can be seen as a surrogate representation of EMGs, and the two are not exactly the same. Hence, mismatches in magnitudes will exist, and this comparison was made to show consistencies in the timing and trends of muscle activations. Furthermore, raw EMG values can vary widely due to electrode placement.
>
> $\textbf{W5, on evaluation on gross movements over fine-motor movements:}$
> fast and dynamic movements such as baseball pitching and golf swings were chosen deliberately to better assess kinetic estimations, as they are more prone to errors leading to unrealistic force and torque estimations.
>
> $\textbf{Q1, How accurate or realistic are the estimations of ground reaction forces and external forces, since the contact is simplified?}$
> $\textbf{What's the inference efficiency?}$
> Ground reaction force (GRFs)  are evaluated in Figs. 3,5. Overall, compared to PhysPT, our median sum of vertical GRFs is much closer to the body weight of the subject. For golf swings and baseball pitching, our 99\% quantiles are also closer to forceplate values reported in biomechanics studies [50, 59]. For gait, our vertical GRF estimates exhibit the characteristic "m" shape. Our method adds about 0.003 seconds to the inference time of the existing kinematics pose estimator used, for a video sequence of 16 frames, on a Titan Xp GPU.

---

### Official Review · Reviewer_i28J · 2025-07-03

**Clarity:** 2
**Significance:** 2
**Originality:** 4
**Rating:** 4
**Confidence:** 3

**Summary:**

This paper presents a method to refine the output of existing forward-kinematics pose regressors (SMPL prediction models) by imposing constraints inspired by muscle models such as torque, limb lengths. (more below)

**Questions:**

L240 says an example consists of 16 frames, but how much real-world time does this correspond to? AMASS has a very high FPS (60-120+ Hz) depending on the subset. I think it is important to know how much motion is captured.

L269 says, the worse performance on 3DPWoc is within reasonable margin compared to SOTA (the best one in the table is CLIFF). But according to L259, CLIFF was used as input to the proposed model. This input-output relationship could be made clear. It's not necessarily a limitation considering other metrics may have improved (Table 3 in supp).

Could you explain the "with alpha-hat" ablation in Table 2 more? I found it hard to understand.

**Ethical Concerns:**

["NO or VERY MINOR ethics concerns only"]

**Final Justification:**

The authors have addressed my concerns, particularly regarding the motion-based comparison, so I would like to increase my rating. I also accept that SMPL joints are positioned at the locations where rotation occurs. While I remain uncertain about making claims regarding faithfulness to real biological structures, I believe the experiments are valuable regardless of the specific wording. I defer to the judgment of the other reviewers.

**Limitations:**

Yes

**Paper Formatting Concerns:**

No formatting concerns

**Quality:**

3

**Strengths And Weaknesses:**

+ I think the proposed novel components are well motivated. It could potentially be used to augment existing inference or evaluation methods.

+ Figure 4 seems to show muscle activations are meaningful.

Some of the muscle-related claims can be misleading, e.g. application to "whole-body musculoskeletal dynamics of athletes" (L55, L58) and biofidelity. SMPL joints labels are not consistent with anatomy (e.g. hips; https://files.is.tue.mpg.de/black/talks/SMPL-made-simple-FAQs.pdf). Some of this paper's core designs (e.g. passive and active torque losses) were inspired by and are analogous to muscles, but the interdisciplinary claims do not seem compelling enough. Also, I feel that simpler alternatives could have been explored and compared against.

Considering the main selling point is biofidelity and physical plausibility, I think more analysis on temporal consistency (e.g. smoothness, jitter, or torque continuity metric) would have been helpful.

CLIFF might be a little dated now. More recent, motion-based approaches such as CoMotion (ICLR 2025) could have been insightful.

Comparison with existing energy-minimization approaches in vision (limb length symmetry constraints, foot skating loss) would have been helpful, e.g. hypothetically, results showing those existing techniques should be used with the proposed ones can be directly helpful to other researchers.

---

> ### Author Rebuttal · Authors · 2025-07-31
>
> We thank you for your thoughtful feedback. We are encouraged you found our method novel, well motivated, compatible with existing methods, and able to infer meaningful muscle activations. We hope our responses to Weakness $\textbf{W2-4}$ below can improve the significance of our paper, and that our answers to $\textbf{W1}$ and Questions $\textbf{Q1-3}$ clear up some of the clarity concerns.
>
> $\textbf{W1(a),  SMPL joints labels are not consistent with anatomy:}$
> in your referenced link, Slide11 says, "SMPL puts the hip joint where the rotation happens," which is what we desire to model biomechanically.
> Slide12 depicts the inconsistency of the H36M hip label (not SMPL), and shows its difference with the SMPL hip joint center.
>
>
> $\textbf{W1(b), interdisciplinary claims not compelling enough, simpler alternatives could have been explored and compared against:}$
> we chose MTGs as a middle-ground between $\textit{\textbf{simple}}$ alternatives to estimate torques that other physics-based human pose estimators (PHPE) currently employ and $\textit{\textbf{complex}}$ muscle models, to support our claim of improving biofidelity over the $\textit{\textbf{simple}}$ alternatives (with results shown in Sec 4.3), while further allowing us to infer human movement at a musculoskeletal level; at the same time, MTGs are compatible with our learning framework and maintains model parsimony by avoiding detailed musculoskeletal geometry and granular joint contact force computations that complex muscle models require (as discussed in L64-74 and L139-145).
> We will expand on the MTG details in Sec B.4 of the Supp. A more complete motivation for MTGs can be found in [Inkol et al, 2020, Muscle torque generators in multibody dynamic simulations of optimal sports performance].
>
>
> $\textbf{W2, more analysis on temporal consistency:}$
> thank you for this suggestion. We will add the following table to further assess temporal consistency.
> Specifically, both our method and PhysPT improve kinematically (ACC, JACC) over CLIFF, while our method further improves kinetically (MTV) over PhysPT.
> Here JACC is the mean absolute joint angle acceleration loss (in degrees/frame\^2), and MTV is the mean torque variation as the mean absolute change in joint torques over consecutive frames (in Newton*metres/frame). Lower is better for both.
> Joint position acceleration loss (ACC) is already reported in Tabs. 3,4 of the Supp.
>
> |        | **H36M** |     |   | **3DPWoc** |      |
> |--------|:--------:|-----|---|:----------:|------|
> |        | JACC     | MTV |   | JACC       | MTV  |
> | CLIFF  | 2.7      |  -  |   | 1.8        | -    |
> | PhysPT | 1.6      | 5.3 |   | 0.84       | 27.0 |
> | ours   | 1.6      | 2.5 |   | 0.95       | 12.1 |
>
> $\textbf{W3, using a more recent kinematic model than CLIFF:}$  thank you for this suggestion. We planned on adding results for CoMotion here, but unfortunately ran out of time. $\textbf{Recap:}$ our goal is a framework that works with most kinematic models, and similar to your suggestion, we have shown consistent results using WHAM in Tab. 4.  $\textbf{The motivation for using CLIFF}$ during inference (in L256-260) was (1) because it produces competitive positional accuracy but lacks in physical plausibility (for which we demonstrated improvements in), and (2) for an even comparison with PhysPT (which uses CLIFF). However, we will add results for additional kinematics estimators to our Supp.
>
>
> $\textbf{W4, incorporating existing energy-minimization approaches in vision:}$
> below are the results trained with an additional $\textbf{skating loss}$ [77] to minimize feet velocity when in contact with the ground:
>
> |               | **H36M** |      |      |         |   | **3DPWoc** |      |      |         |
> |---------------|:--------:|------|------|---------|---|:----------:|------|------|---------|
> |               | MJE      | resF | FS   | float\% |   | MJE        | resF | FS   | float\% |
> | MusclePose    | 48.3     | 0.08 | 37.2 | 3.0     |   | 27.6       | 0.3  | 12.8 | 4.7     |
> | w/ skate loss |   51.2   | 0.44 | 27.3 | 82.7    |   | 31.2       | 2.6  | 16.8 | 83.5    |
>
> ($\textit{MJE}$: mean joint error in mm. $\textit{resF}$: residual force scaled by body weight. $\textit{FS}$: foot skating, average displacement in mm of vertices in contact with the ground in consecutive frames. $\textit{float\\%}$: percentage of frames where the human is more than 10 mm off the ground. Lower is better for all.)
>
> Although we see some reduction in foot skating (FS) in H36M for this experiment, the other metrics are all worse off. While these additional kinematic constraints are indeed compatible with our method, and this experiment could be further fine-tuned, they may not be needed for PHPE, as they are already incorporated into the kinetics loss -- these kinematic artifacts may lead to high residual forces or unrealistic ground reaction forces.
>
> For $\textbf{limb length symmetry}$, we did not run additional experiments, as it did not seem like a significant issue for SMPL-based human pose estimators. The limb length symmetry error was about 4.5-4.7 mm for our method, PhysPT, and CLIFF, which seemed anatomically reasonable.
>
>
>
> $\textbf{Q1, amount of motion captured:}$
> during training, we downsampled AMASS to 10 fps (1.6 seconds of real world time for a sequence).
>
> $\textbf{Q2(a), input-output relationship:} $
> as discussed in $\textbf{W3}$, any kinematic model could be used as input, and CLIFF was used during evaluation for even comparison with PhysPT and to better show improvements in physical plausibility. We will clarify better in the paper.
>
> $\textbf{Q2(b), explanation for ``kinematics-kinetics trade-off" in 3DPWoc:}$
> since this dataset was captured with a moving camera, the root degrees of freedoms are harder to estimate, potentially leading to higher residual forces, which the model may try to reduce (lower kinetic error) by estimating a set of local joint kinematics slightly different from the original motion (higher kinematic error).
>
>
>
> $\textbf{Q3, $\hat\alpha$ ablation:}$
> we added this ablation as an additional way to compare with existing methods that estimate joint torques as some linear combination of the joint's kinematics $\mathcal{K} = \\{ \text{joint angles } q, \text{angular velocity } \dot{q},... \\}$,
>
> $
> \text{ours: }\\; \tau_{MTG} = \alpha \cdot \tau_{active}(q, \dot{q}) + \tau_{passive}(q)
> $
>
> $
> \text{existing: }\\; \tau = b_0 + b_1 \dot{q} + b_2 q + b_3 F(\mathcal{K}),\qquad
> $
> (for learned coefficients $b$'s, and some parametric $F$)
>
> Since the muscle signal $\alpha$ can be interpreted as an ``effort level" between 0 and 1, for the ablation we estimated it as $\hat{\alpha} = (\dot{q} /\dot{q}_{limit}) + \epsilon$,
>
> where $\dot{q}_{limit}$ is the joint's anatomical angular velocity limit, and $\epsilon$ is a predicted offset term.
>
> $\textbf{Results:}$ the worsened results from this initialization could be due to its over-reliance on the initial kinematics estimates.

---

> ### Author Response · Authors · 2025-08-04
> **Re: W3, using CoMotion as the kinematics estimator**
>
> $\textbf{Re: W3, using CoMotion as the kinematics estimator}$: below are the results with CoMotion as the kinematics estimator. Despite the different initial kinematic estimates, ours consistently results in lower residual forces, which is the main focus of our paper.
>
> |                          | **H36M**     |        |   | **3DPWoc** |      |
> |-------------------------:|:------------:|:------:|---|:----------:|:----:|
> |                          |          MJE |   resF |   | PJE        | resF |
> | CoMotion                 |         52.3 |    -   |   |       30.7 |   -  |
> | PhysPT (w/ CoMotion)     |         59.2 |    0.5 |   |       32.7 |  0.9 |
> | ours (w/ CoMotion)       |         56.8 |    0.1 |   |       32.6 |  0.2 |
> |                          |              |        |   |            |      |
> | CLIFF                    |         46.5 |    -   |   |       24.0 |   -  |
> | PhysPT (w/ CLIFF)        |         50.6 |    0.4 |   |       25.9 |  0.9 |
> | ours (w/ CLIFF)          |         48.4 |    0.1 |   |       27.6 |  0.3 |

---

### Comment · Area_Chair_TxEB · 2025-08-03
**Author-Reviewer Discussion**

Dear Reviewers,

The author-reviewer discussion period is now open and will continue until August 6. Please review the authors’ rebuttal to determine whether it adequately addresses your concerns.  If you have further questions or comments, engage with the authors by acknowledging that you’ve read their response and providing additional feedback as needed

Sincerely,

Your AC

---

### Note · Authors · 2025-08-12

Dear reviewers and AC,

Thank you. We really appreciate you for taking time out of your busy schedules to review our paper.

We are encouraged that you found our method novel, well-motivated, compatible with existing methods [i28J], and impactful as an improvement over existing regression-based physics pose estimators, with better kinetics estimates [m1ia, xJ9Z, TsEp] and meaningful muscle predictions [i28J].

We are also grateful for all of your constructive feedback, and feel that the revised version of our paper is now more complete, with additional evaluations and ablations, and improved clarity of technical details.

Thanks,\
Authors

---

### Decision · Program_Chairs · 2025-09-17

**Decision:**

Accept (poster)

**Comment:**

This paper introduces a 3D human pose estimator that integrates muscle-dynamics modeling and detailed anthropometrics to simultaneously predict human kinematics, kinetics, muscle signals, and anthropometrics from monocular video. Experiments on benchmark datasets demonstrate improved accuracy and bio-fidelity compared to purely kinematics-based models.  The paper received four reviews: one accept and three borderline accepts. Reviewers acknowledge the novelty of explicitly combining muscle modeling, kinematic modeling, and anthropometrics to produce biologically faithful 3D body pose estimation. They also recognize the experimental results showing improved performance over purely kinematic methods.   However, reviewers raise several concerns. Major issues include the evaluation setting being limited to foot–ground contact, the absence of quantitative evaluation of estimated muscle signals, evaluation restricted to gross body motion, questions about the biological faithfulness of the estimated poses, inadequate evaluation metrics, and the lack of computational efficiency analysis.  The authors provide an extensive and detailed rebuttal with additional experimental results, which effectively addresses most of the reviewers’ concerns.